# HyperTree Planning: Enhancing LLM Reasoning via Hierarchical Thinking

**Runquan Gui** [1 2]   **Zhihai Wang** [1 3]   **Jie Wang** [1] [†]   **Chi Ma** [1]   **Huiling Zhen** [3]   **Mingxuan Yuan** [3]   **Jianye Hao** [3 4]
**Defu Lian** [5]   **Enhong Chen** [5]   **Feng Wu** [1]

## Abstract

Recent advancements have significantly enhanced the performance of large language models (LLMs) in tackling complex reasoning tasks, achieving notable success in domains like mathematical and logical reasoning. However, these methods encounter challenges with complex planning tasks, primarily due to extended reasoning steps, diverse constraints, and the challenge of handling multiple distinct sub-tasks. To address these challenges, we propose **H**yper**T**ree **P**lanning (HTP), a novel reasoning paradigm that constructs hypertree-structured planning outlines for effective planning. The hypertree structure enables LLMs to engage in hierarchical thinking by flexibly employing the divide-and-conquer strategy, effectively breaking down intricate reasoning steps, accommodating diverse constraints, and managing multiple distinct sub-tasks in a well-organized manner. We further introduce an autonomous planning framework that completes the planning process by iteratively refining and expanding the hypertree-structured planning outlines. Experiments demonstrate the effectiveness of HTP, achieving state-of-the-art accuracy on the TravelPlanner benchmark with Gemini-1.5-Pro, resulting in a $3.6\times$ performance improvement over o1-preview.

## 1. Introduction

Planning has long been recognized as a core skill for intelligent agents and a key benchmark for evaluating the cognitive capabilities of large language models (LLMs) (Zhu et al.,

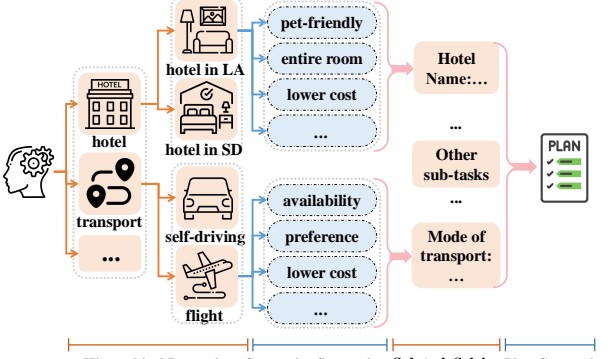

*Figure 1.* An example of hierarchical thinking, demonstrating its ability to decompose complex reasoning chains into manageable components, effectively handle diverse constraints, and systematically manage multiple distinct sub-tasks.

2022; Shum et al., 2023). Unlike mathematical or logical reasoning tasks, which have clear, definitive answers (like numbers or specific nouns), planning tasks require a feasible plan consisting of multiple interdependent components. For example, a travel plan involves selecting hotels, booking flights, choosing restaurants, and deciding on attractions, each of which can be further broken down with various constraints. Hotel selection, for instance, may involve factors like house rules, room types, budget limitations, and more. Therefore, completing complex planning tasks typically requires reasoning over extended steps, decision-making under diverse complex constraints, and generating comprehensive and feasible solutions, which presents unique challenges for existing LLMs (Ju et al., 2024; Xie et al., 2024; Huang et al., 2023; Valmeekam et al., 2024b).

A series of works have been proposed to enhance the planning capabilities of LLMs. In-context learning approaches, by providing curated demonstration examples before inference, enable models to improve their planning ability through analogical reasoning (Sprague et al., 2024; Chen & Li, 2024). An innovative breakthrough is Chain-of-Thought (CoT) prompting (Wei et al., 2022; Kojima et al., 2022), which extends the reasoning process from a simple input-output format into a multi-step chain of reasoning, enabling models to think step-by-step. Building on this, Tree-of-Thought (ToT) prompting (Yao et al., 2024) takes it a step

---

[1]MoE Key Laboratory of Brain-inspired Intelligent Perception and Cognition, University of Science and Technology of China [2]<rqgui@mail.ustc.edu.cn> [3]Noah's Ark Lab, Huawei Technologies [4]College of Intelligence and Computing, Tianjin University [5]State Key Laboratory of Cognitive Intelligence & University of Science and Technology of China. Correspondence to: Jie Wang <jiewangx@ustc.edu.cn>.

*Proceedings of the $42^{nd}$ International Conference on Machine Learning*, Vancouver, Canada. PMLR 267, 2025. Copyright 2025 by the author(s).

further by transforming the linear reasoning process into a branching structure, allowing LLMs to reason along multiple paths and systematically explore the search space. Recent planning methods have increasingly explored innovations in agent systems, where specialized agents collaborate through structured processes to tackle complex planning tasks, further extending the scope and effectiveness of planning solutions (Yuan et al., 2024; Zhang et al., 2024a).

Despite these advances, existing planning methods exhibit several limitations. First, existing reasoning paradigms primarily focus on mathematical or logical reasoning tasks, making them ill-suited for planning problems and their unique challenges. Second, the performance of LLMs relying on in-context learning is highly dependent on the quality of the provided examples, requiring significant human expertise and constraining generalization ability (Dong et al., 2022; Zhao et al., 2021; Lu et al., 2021). Although various methods have been proposed to address these limitations (Liu et al., 2021; Tanwar et al., 2023; Qin et al., 2023), the discrepancy between the example and the actual query remains a fundamentally unresolved challenge. Third, current autonomous agent methods consistently rely on human-designed interventions, such as requiring the manual creation of distinct personas and task-specific descriptions for each agent, which hinders generalization across diverse tasks (Hong et al., 2023; Liang et al., 2023).

To address these challenges, this paper proposes **HyperTree Planning** (HTP), a novel reasoning paradigm that constructs hypertree-structured planning outlines for effective planning [1]. We observe that humans naturally approach planning tasks by flexibly and hierarchically employing the divide-and-conquer strategy. They continuously break down a task into smaller, more manageable subtasks, address each component individually, and synthesize the results to create the final plan. This cognitive process, known as hierarchical thinking, is illustrated in Figure 1. Inspired by this, HTP extends the ToT framework by incorporating the hypertree structure proposed in (Lample et al., 2022). In this enhanced framework, each edge connects a parent node to a set of child nodes, providing an intuitive structural foundation for the divide-and-conquer mechanism. By flexibly applying this strategy across multiple layers, our planning outline effectively implements hierarchical thinking tailored to the original query. Furthermore, we design an autonomous planning framework that seamlessly integrates these planning outlines into self-guided planning and plan generation processes, ultimately producing high-quality plans.

Specifically, we begin by modeling step-by-step reasoning, multi-path inference, and the divide-and-conquer within the hypertree structure. Next, we introduce the top-down hypertree construction algorithm, which systematically applies these strategies to build the hypertree-structured planning outline. Following this, we refine and expand the planning outline iteratively, ultimately generating the final plan. Experimental results demonstrate that HTP, leveraging GPT-4 or Gemini-1.5-Pro as its backbone, achieves significant performance improvements across multiple complex planning benchmarks, outperforming state-of-the-art agent methods, planning strategies, and closed-source models. The core contributions are summarized as follows:

- **HyperTree Reasoning Paradigm**: We propose HTP, a novel hypertree reasoning paradigm, which, to the best of our knowledge, is the first to use a hypertree structure to model the reasoning process, empowering LLMs to perform hierarchical thinking.

- **Novel Planning Framework**: We present a fully autonomous planning framework that leverages task-specific planning outlines to self-guide the planning process, generalizing effectively to diverse tasks without relying on manually crafted examples.

- **Superior Performance**: HTP significantly outperforms existing methods on complex planning benchmarks, achieving 36.1% accuracy on TravelPlanner with Gemini-1.5-Pro, substantially outperforming o1-preview (10.0%).

## 2. Related Work

**Reasoning Paradigms** Owing to its parameter-free nature, the prompting paradigm has garnered significant attention and emerged as a promising approach to unlocking the reasoning potential of LLMs (Zhou et al., 2024; Edelman et al., 2024; Jeon et al., 2024; Lin & Lee, 2024). Few-shot prompting, in particular, provides curated demonstration examples before inference, enabling LLMs to perform reasoning tasks by leveraging analogical examples (Wei et al., 2022; Sprague et al., 2024; Chen & Li, 2024). However, their reasoning performance heavily depends on the quality of the provided examples, requiring substantial human expertise and limiting their generalization ability (Dong et al., 2022; Zhao et al., 2021; Lu et al., 2021). In response to this limitation, HiAR-ICL (Wu et al., 2024) proposes replacing demonstration examples with reasoning patterns formed by permutations of five fixed actions. However, the guidance from fixed elements is quite limited, and due to the chain-like structure of the patterns, they cannot fully support the hierarchical thinking needed for planning.

Starting from Chain-of-Thought (CoT) (Wei et al., 2022), a series of techniques have been developed to improve the reasoning capabilities of LLMs by introducing novel reasoning paradigms. Zero-shot-CoT (Kojima et al., 2022)

---

[1] A hypergraph is a graph where an edge leads to a set of nodes. A hypertree is a hypergraph without cycles. More formal definitions can be found in Section 3.1.

enables reasoning step generation without the need for examples. Departing from the linear, left-to-right chain-like structure, a significant body of work, exemplified by ToT (Yao et al., 2024), has extended reasoning paradigms into tree-based frameworks to broaden potential search spaces, subsequently employing diverse tree search algorithms to improve reasoning precision (Hao et al., 2023; Wang et al., 2024c; Qi et al., 2024; Zhang et al., 2024b; Feng et al., 2023; Putta et al., 2024). However, these tree search-based methods focus on optimizing the CoT path through trials, lacking fundamental innovations in the reasoning process itself (Zhang et al., 2024c). Plan-and-Solve (Wang et al., 2023a) requires LLMs to devise a plan that breaks the task into smaller subtasks, and then carry out the subtasks according to the plan. Similarly, Skeleton-of-Thought (Ning et al., 2024) first generates a skeleton answer outline, and then completes content in parallel to reduce generation latency. These methods demonstrate some divide-and-conquer capability, but they lack technical innovation in constructing the outline and are limited to a single level of task decomposition due to insufficient support from reasoning structures.

**Planning Agents** Autonomous agents are systems designed to execute diverse tasks through self-directed actions (Wang et al., 2024b), with planning serving as a fundamental function (Valmeekam et al., 2024a; Xie et al., 2024; Zheng et al., 2024; Zhang et al., 2024a). Methods such as ReAct (Yao et al., 2022) and RAP (Hao et al., 2023) formalize planning as a Markov decision process, allowing agents to dynamically reason and act by evaluating intermediate states and iteratively making decisions. Recent studies have leveraged multi-agent systems to improve the planning capabilities in specific tasks (Chen et al., 2023a;c; Zhang et al., 2024a). However, these approaches often rely heavily on human-designed interventions, including the creation of tailored personas and task-specific descriptions for each agent. EvoAgent (Yuan et al., 2024) utilizes evolutionary algorithms to automate agent role generation, but its ability to address diverse constraints remains limited. Another promising direction entails integrating LLMs with external planning tools (Dagan et al., 2023; Guan et al., 2023; Yang et al., 2023; Ju et al., 2024; Lee et al., 2025). While these hybrid approaches can achieve high accuracy, the construction of feasible solvers or evaluators remains challenging, with high engineering overhead, design costs, and a lack of generalization across different datasets.

## 3. Preliminary

In this section, we first illustrate the problem statement in Section 3.1, and subsequently define the concepts related to hypertrees in Section 3.2, drawing from the definitions of hypertree proof search (Lample et al., 2022).

### 3.1. Problem Statement

Given a planning query $q$ and a pre-trained LLM $\pi_\theta$, the problem-solving process can typically be divided into two phases: planning and plan generation. The planning phase involves generating intermediate reasoning results, formalized as $\mathcal{C} = \pi_\theta(\Phi(q))$, where $\Phi$ denotes the predefined instructions. Building on this, the plan generation phase refines $\mathcal{C}$ into the final solution, expressed as $\mathcal{P} = \pi_\theta(\Phi(\mathcal{C}))$.

In the planning phase, a common approach is to frame it as a multi-step thinking process. Specifically, we guide the large model to generate a sequence of reasoning steps starting from $q$, which results in $\mathcal{C}$ that completes the phase. In detail, suppose there are $N$ intermediate reasoning steps $s_{0...N} = [s_0, s_1, s_2, ..., s_N]$, where $s_0 = q$ and $s_N = \mathcal{C}$. At each time step $n$, the model receives a state $S_{n-1}$, which consists of the original input $q$ and the preceding reasoning steps $(s_1, s_2, ..., s_{n-1})$, and generates the current reasoning step $s_n = \pi_\theta(\Phi(S_{n-1}))$. The sequence $s_{1...N} = [s_1, s_2, ..., s_N]$ forms a complete chain-like structure of step-by-step reasoning (Wei et al., 2022).

Building on the chain-like structure, ToT (Yao et al., 2024) extends a tree structure to explore different choices and their outcomes. Given the current state $S_{n-1}$, the LLM generates multiple feasible steps $\{s_n^{(1)}, \ldots, s_n^{(m)}\} = \pi_\theta(\Phi(S_{n-1}))$, each of which remains independent in subsequent expansions. This branching allows the model to explore the search space and results in a tree $\mathbf{T}$, from which a single leaf is ultimately selected as the result $\mathcal{C}$.

### 3.2. HyperTree Definition

Formally, let $G$ represent a set of nodes, and $R$ a set of rules. A hypertree library is defined as a tuple $L = (G, q, R)$, where $q \in G$ serves as the root node. Each rule $r \in R$ is a function $r : g \mapsto c$, where $g \in G$ is referred to as the start node, and $c \subseteq G$ represents the corresponding set of child nodes. Here, $\mathcal{P}(G)$ denotes the power set of $G$. To ensure acyclicity, no sequence $g_0, g_1, \ldots, g_\ell = g_0$ ($\ell > 0$) is allowed, where $g_{i+1}$ is a child of $g_i$ for all $0 \le i < \ell$. The set of divisible nodes is defined as:

$$D = \{d \in G \mid \exists r \in R \text{ such that } d \text{ is the start node of } r\}.$$

Using the hypertree library $L$, a generating hypertree $\mathbf{H}$ is defined as a hypertree satisfying the following properties:

1. Every leaf node of $\mathbf{H}$ belongs to $G$.

2. Every non-leaf node of $\mathbf{H}$ belongs to $D$.

3. For any non-leaf node $g$ with a branch of child nodes $c$, there exists a rule $r \in R$ such that $r : g \mapsto c$.

A special case of $\mathbf{H}$, referred to as a hyperchain $\mathbf{C}$, is a hypertree with no branching. In a hyperchain, each non-leaf node $g_i \in \mathbf{C}$ generates its child nodes using the same rule $r$. Examples of $L$ can be found in Appendix E.1.

## Tree Planning    HyperTree Planning (HTP)

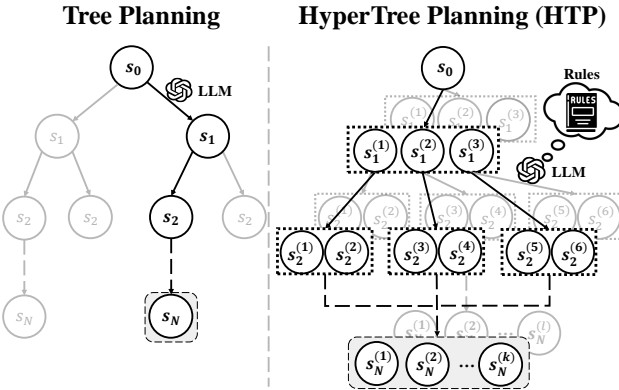

*Figure 2.* An overview of HyperTree Planning (HTP). Compared to previous tree planning methods such as ToT (Yao et al., 2024) and RAP (Hao et al., 2023), HTP introduces structural innovations that enable each edge to connect multiple child nodes, making it suitable for a divide-and-conquer strategy.

## 4. Methodology

**Overview of HTP** In this section, we introduce HTP in detail and illustrate the entire process in Figure 3. We begin by modeling the reasoning path as a hypertree and, based on this framework, introduce a top-down hypertree construction algorithm. This algorithm generates a hypertree-structured planning outline $\mathcal{O}$ tailored to the specific query $q$. Guided by $\mathcal{O}$, the self-guided planning process systematically solves the corresponding sub-tasks, completes the planning process, and derives $\mathcal{C}$. Finally, the plan generation process integrates the results into a comprehensive final plan $\mathcal{P}$.

### 4.1. From Tree Planning to HyperTree Planning

Building on the concept of hypertrees, HTP introduces a hypertree-structured reasoning paradigm. For a real-world planning dataset, let $R$ denote a fixed set of rules derived from the descriptions of the dataset, while $G$ represents reasoning steps expressed in natural language. As shown in Figure 2, given the current state $S_{n-1}$, suppose a divisible leaf node $s_{n-1}$ is selected. By definition of a hypertree, the next reasoning steps following $s_{n-1}$ can be represented as:

$$\big\{\{s_n^{(1,1)}, \ldots, s_n^{(1,k_1)}\}, \ldots, \{s_n^{(m,1)}, \ldots, s_n^{(m,k_m)}\}\big\},$$

where $m$ represents the number of branches, and $k_i$ denotes the number of child nodes within the $i$-th branch. Similar to tree planning methods, since the branches are independent of one another, we ultimately select a hyperchain $\mathbf{C}$ from $\mathbf{H}$, which contains no branches. Once the hyperchain $\mathbf{C}$ is fully expanded, the set of all leaf nodes collectively serves as the outcome $\mathcal{C}$ of the planning phase, in contrast to previous methods that select only a single leaf node.

Similar to tree planning, we continue to use edges to represent the connections underlying sequential reasoning in the hypertree structure. Since all child nodes within the same branch share the same edge with the parent node,

---

**Algorithm 1** Top-down HyperTree Construction Algorithm
**Input:** rules $R$, query $q$, LLM $\pi_\theta$, reasoning depth $K$, expansion width $W$
Convert divisible set: $D \leftarrow \text{Convert}(R)$
Initialize hypertree: $\mathbf{H} \leftarrow q$
**for** $d \leftarrow 1$ **to** $K$ **do**
    Extract hyperchains: $\{\mathbf{C}_1, \ldots, \mathbf{C}_m\} \leftarrow \text{Map}(\mathbf{H})$
    **if** $m > W$ **then**
        Filter hyperchains: $\{\mathbf{C}_1, \ldots, \mathbf{C}_W\} \leftarrow \pi_\theta(\mathbf{H})$
    **end**
    **for** $i \leftarrow 1$ **to** $\min(W, m)$ **do**
        Extract divisible nodes: $g_1, \ldots, g_{n_i} \leftarrow \pi_\theta(\mathbf{C}_i, D)$
        Select node: $g_i^* \leftarrow \pi_\theta(q, \mathbf{H}, g_1, \ldots, g_{n_i}))$
        Retrieve rules: $r_1, \ldots, r_P \leftarrow \pi_\theta(R, g_i^*)$
        **for** $p \leftarrow 1$ **to** $P$ **do**
            Expand nodes: $\{s_i^p\} \leftarrow \pi_\theta(q, \mathbf{C}_i, g_i^*, r_p))$
            $\mathbf{H} \leftarrow \text{AttachNodes}(\{s_i^p\}, \mathbf{H}, g_i^*)$
            $p \leftarrow p + 1$
        **end**
        $i \leftarrow i + 1$
    **end**
    $d \leftarrow d + 1$
**end**
Select the optimal hyperchain: $\mathbf{C}^* \leftarrow \pi_\theta(\mathbf{H})$
**return** Planning Outline $\mathcal{O} \leftarrow \mathbf{C}^*$

---

the structure inherently reflects a divide-and-conquer strategy. Additionally, due to the inherent flexibility of the hypertree structure, any divisible child node $s_n^{(i,j)}$ generated from $s_{n-1}$ can be further expanded through successive divide-and-conquer steps. While the chain structure equips LLMs with step-by-step reasoning and the tree structure enables exploration across multiple paths, the hypertree structure uniquely introduces a capability we term as **hierarchical thinking**—a multi-level divide-and-conquer approach that facilitates deeper and more organized reasoning. Through hierarchical thinking, each path ultimately evolves to address distinct subtasks, with the final outcomes $\{s_N^{(1)}, s_N^{(2)}, \ldots, s_N^{(k)}\}$ representing the solutions to the respective subtasks.

### 4.2. Top-down Hypertree Construction Algorithm

The core of HTP is to generate a hypertree-structured planning outline based on the given rules $R$. To the best of our knowledge, no existing hypertree construction algorithm has been designed for reasoning tasks. In this context, we propose a top-down hypertree construction algorithm that starts from the root node and progressively builds a hypertree. Specifically, the algorithm unfolds in four stages: *selection*, *expansion*, *construction*, and *decision*.

*(1) Selection.* Suppose the current hypertree is $\mathbf{H}$, which can naturally be mapped to a series of hyperchains $\{\mathbf{C}_1, \ldots, \mathbf{C}_m\}$ based on its distinct branches, where $m$ represents the total number of hyperchains. This phase aims to identify

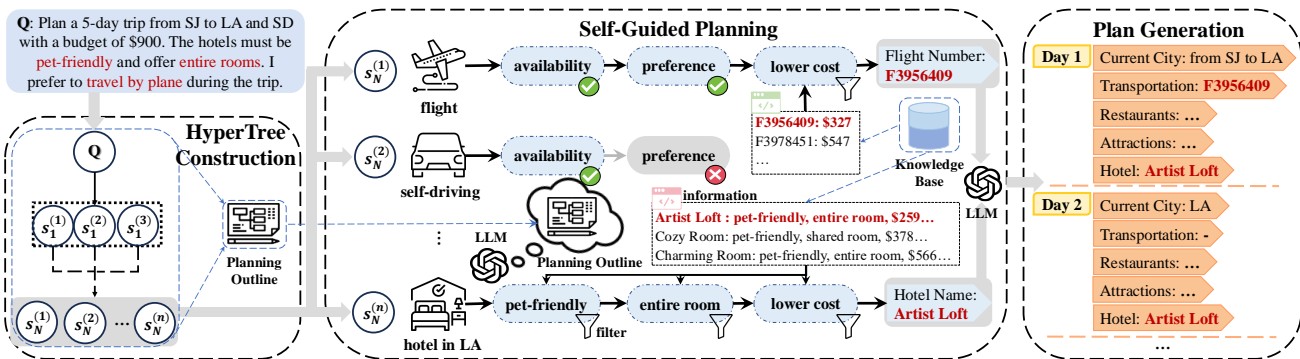

*Figure 3.* Flowchart of HTP, which consists of three parts: (1) HyperTree Construction; (2) Self-Guided Planning; (3) Plan Generation.

the appropriate leaf nodes of $\mathbf{H}$ for subsequent expansion, which involves two main steps: first, selecting the optimal hyperchains from $\mathbf{H}$, and then determining the most suitable leaf node within each chosen hyperchain.

To effectively select the optimal hyperchains from $\mathbf{H}$ and manage their number, inspired by the tree-structured methods for limiting width, we adopt three strategies: a width-based pruning method, which restricts the total number of branches; a probability-based pruning method, where hyperchains with low confidence probabilities—generated by the LLM during branching—are eliminated; and an LLM-guided evaluation method, which leverages the LLM to filter and assess candidate hyperchains.

Given a selected hyperchain $\mathbf{C}$, the next step is to identify the most suitable leaf node. Since only divisible nodes can be expanded further, we start by extracting the divisible leaf nodes from $\mathbf{C}$. Determining whether a node is divisible is straightforward, as the starting nodes under the given rules $R$ typically follow a specific format, which is detailed in Appendix E.1. When selecting a leaf node from the divisible leaf nodes, traditional methods like UCT (Kocsis & Szepesvári, 2006), PUCT (Silver et al., 2017), and Regularized Policy (Grill et al., 2020) are not suitable. These methods are designed for scenarios where paths perform the same task, balancing exploration and exploitation. However, in our setting, paths are to address distinct subtasks, making these approaches ineffective. Furthermore, simple heuristics fail to capture the broader context of the entire hyperchain, which is critical for making appropriate node selections. Therefore, we turn to LLMs, which can analyze the rules $R$ and the structure of the current hyperchain $\mathbf{C}$ to identify the most promising leaf node for expansion.

*(2) Expansion.* During the expansion phase, given the selected node $g$, we first retrieve the set $S = \{r \in R \mid g$ is the start node of $r\}$, which includes all the rules where $g$ serves as the starting node. We then sample $P$ rules $r_1, \ldots, r_P$ from the set $S$. For each sampled rule $r_p$, we generate the corresponding child nodes $\{s^{(p,1)}, \ldots, s^{(p,k_p)}\} = \pi_\theta(\Phi(q, \mathbf{C}, g, r_p))$, which are collectively treated as a single

branch and added to $\mathbf{C}$ as part of the expansion for node $g$.

*(3) Construction.* The construction process for a single hyperchain $\mathbf{C}$ involves iteratively performing selection and expansion, starting from the root $q$, until either all leaf nodes become indivisible or the iteration limit is reached.

*(4) Decision.* Once the hypertree $\mathbf{H}$ is fully constructed, the decision-making process takes place. During this phase, LLMs are prompted to identify the optimal hyperchain within $\mathbf{H}$, which will then serve as the final planning outline $\mathcal{O}$ for the hypertree construction algorithm.

The algorithm generates a hypertree-structured planning outline automatically for any query, requiring no human intervention. With features such as task decomposition, multi-path exploration, and self-evaluation, it adapts seamlessly to diverse planning frameworks without prerequisites. The entire process is illustrated in Figure 3, and the details of our algorithm are provided in Algorithm 1.

### 4.3. Self-guided Planning and Plan Generation

After executing the top-down hypertree construction algorithm, we obtain a planning outline $\mathcal{O}$ for the query $q$. While this outline is structurally clear and comprehensive, it lacks the detailed content required for a fully developed plan. This limitation arises from two main factors. First, certain planning tasks like TravelPlanner rely on external knowledge—such as a list of tourist attractions—to create a complete plan, and the knowledge base is not included during the hypertree construction process. Second, the leaf nodes of the planning outline generally break the task down into subtasks, but further in-depth reasoning is needed to address these subtasks and generate their corresponding solutions.

To address these gaps, we provide the planning outline $\mathcal{O}$, along with any available knowledge base $\mathcal{K}$, to the LLM $\pi_\theta$. The self-guided process expands and enriches the outline with detailed content while maintaining its hierarchical structure. Specifically, for non-leaf nodes in $\mathcal{O}$, we leverage $\mathcal{K}$ to refine them. For leaf nodes in $\mathcal{O}$, we facilitate further growth of the hypertree by iteratively expanding the nodes

*Table 1.* Main results of different LLMs and planning strategies across three planning benchmarks. DR, CPR, HCPR, and SR represent the Delivery Rate, Commonsense Pass Rate, Hard Constraint Pass Rate, and Success Rate, respectively, while **Blocks.**, **Mys.** and **Trip.** denote the Blocksworld, Mystery Blocksworld and Trip Planning tasks, respectively. The best results are **bolded**, and the second best ones are underlined. Our method, HTP consistently achieves the best performance across models and datasets.

| MODEL | SETTING | TravelPlanner | | | | | | Blocks. | Mys. | Trip. |
| | | DR | CPR | | HCPR | | SR | SR | SR | SR |
| | | | Micro | Macro | Micro | Macro | | | | |
| GPT-4o | CoT | 100 | 80.4 | 18.9 | 46.9 | 22.2 | 4.4 | 35.5 | 0.0 | 15.6 |
| | ToT | 100 | 75.8 | 16.7 | 44.0 | 21.1 | 3.9 | 37.5 | 0.67 | 18.1 |
| | LATS | 100 | 78.3 | 16.7 | 44.8 | 21.1 | 3.9 | 40.2 | 1.7 | 20.6 |
| | One-shot | 100 | 76.4 | 17.2 | 43.3 | 20.6 | 3.9 | 28.2 | 0.67 | 3.7 |
| | HiAR-ICL | 100 | 81.2 | 20.6 | 47.4 | 22.8 | 6.7 | 36.5 | 1.8 | 12.8 |
| | RAP | 100 | 81.0 | 20.6 | **50.5** | 25.6 | 6.7 | 41.2 | 1.8 | 12.5 |
| | EvoAgent | 100 | 81.5 | 21.1 | 31.2 | 18.9 | 7.2 | 34.0 | 0.17 | 13.4 |
| | HTP | 100 | **87.2** | **37.8** | 44.3 | **32.2** | **20.0** | **54.7** | **8.7** | **36.9** |
| Gemini-1.5-Pro | CoT | 100 | 81.4 | 17.2 | 44.5 | 20.5 | 5.6 | 26.7 | 0.0 | 34.7 |
| | ToT | 100 | 79.0 | 18.9 | 46.7 | 21.7 | 6.1 | 31.0 | 0.0 | 36.6 |
| | LATS | 100 | 78.8 | 17.8 | 44.3 | 21.7 | 6.1 | 38.2 | 2.2 | 37.2 |
| | One-shot | 100 | 80.3 | 18.9 | 47.1 | 22.2 | 4.4 | 16.8 | 0.67 | 32.2 |
| | HiAR-ICL | 100 | 81.9 | 20.6 | 50.7 | 26.0 | 7.2 | 24.0 | 1.8 | 34.4 |
| | RAP | 100 | 80.7 | 21.7 | 51.9 | 26.7 | 7.9 | 37.7 | 4.2 | 35.1 |
| | EvoAgent | 100 | 82.6 | 21.7 | 34.5 | 20.5 | 8.9 | 25.3 | 0.68 | 35.6 |
| | HTP | 100 | **91.1** | **55.0** | **62.6** | **49.1** | **36.1** | **67.2** | **19.2** | **42.8** |
| GPT-3.5-turbo | CoT | 100 | 61.0 | 2.8 | 10.0 | 3.3 | 0.0 | 6.7 | 0.0 | 8.8 |
| | ToT | 100 | 57.7 | 2.8 | 8.3 | 2.7 | 0.0 | 13.2 | 0.0 | 10.0 |
| | LATS | 99.4 | 64.4 | 1.7 | 3.1 | 2.7 | 0.0 | 14.2 | 0.0 | 12.5 |
| | One-shot | 100 | 57.8 | 3.9 | 8.6 | 3.3 | 0.0 | 7.8 | 0.0 | 7.2 |
| | HiAR-ICL | 100 | 62.8 | 1.7 | 3.3 | 1.1 | 0.6 | 16.5 | 0.16 | 9.4 |
| | RAP | 100 | **66.3** | 6.7 | 10.7 | 3.3 | 1.1 | 16.7 | 0.16 | 8.8 |
| | EvoAgent | 100 | 64.2 | **7.8** | **11.0** | **4.4** | 1.1 | 12.2 | 0.0 | 7.5 |
| | HTP | 100 | 55.0 | 6.7 | 6.2 | 3.3 | **1.7** | **27.2** | **0.67** | **19.7** |

to address the corresponding subtasks, as shown in Figure 3. Through this self-guided approach, $\pi_\theta$ receives precise guidance tailored to the specific query, empowering it to generate comprehensive and detailed planning outcomes $\mathcal{C}$ that adhere to the specific constraints. Finally, we execute the plan generation process, converting the planning outcomes $\mathcal{C}$ into the required final solution as $\mathcal{P} = \pi_\theta(\Phi(\mathcal{C}))$.

# 5. Experiments

## 5.1. Setups

**Benchmarks** To evaluate the effectiveness of our method, we select three of the most challenging planning datasets: Travel Planner (Xie et al., 2024), PlanBench (Valmeekam et al., 2024a) and Natural Plan (Zheng et al., 2024).

**1) TravelPlanner** is a planning benchmark focused on travel planning, aiming to find an itinerary that satisfies diverse constraints regarding flights, accommodations, and other travel arrangements. In this study, we select the validation set for evaluation, which contains 180 queries and is divided into 9 groups based on difficulty levels (easy, medium and hard) and trip durations (3, 5, and 7 days).

**2) PlanBench** is a benchmark suite that includes domains from the International Planning Competition (IPC, 1998). In this study, we focus on Blocksworld, a commonsense domain centered around stacking blocks on a table, and Mystery Blocksworld, an obfuscated version of the same domain. Each task contains 600 instances.

**3) Natural Plan** is a realistic natural language planning benchmark designed for itinerary creation under specific constraints, simulating real-world planning challenges. In this study, we focus on the Trip Planning task within the benchmark, which evaluates the ability to generate itineraries based on varying requirements. The dataset contains 1,600 queries, divided into eight difficulty levels based on the number of cities involved, ranging from 3 to 10.

**Baselines** We evaluate HTP against four strong baseline categories: (1) planning strategies, including CoT (Kojima et al., 2022), ToT (Yao et al., 2024) and LATS (Zhou et al., 2023); (2) in-context learning methods, including one-shot learning and HiAR-ICL (Wu et al., 2024); (3) agent methods, including EvoAgent (Yuan et al., 2024), RAP (Hao et al.,

2023) and MTP (Zhang et al., 2024a) [2]; (4) powerful LLMs, including o1-preview (OpenAI, 2024b), o1-mini, LLaMA-3.1-405B (AI, 2024), LLaMA-3.1-70B (Dubey et al., 2024), GPT-4 (Achiam et al., 2023), GPT-4o (OpenAI, 2024a), Gemini-1.5-Pro (DeepMind, 2024), Claude 3.5 (Anthropic, 2024a) and Claude 3 (Anthropic, 2024b). We provide detailed descriptions of the baseline methods in Appendix B.1.

**Evaluation Metrics** For all benchmarks, we follow the evaluation metrics specified in the original settings. Specifically, for the TravelPlanner task, we report the delivery rate, commonsense rate, hard constraint rate and success rate. For both the Blocksworld and Mystery Blocksworld tasks, we utilize a plan executor to execute the plans, verify their correctness, and report the success rate. In the Trip Planning task, since the correct plan is unique, we calculate the success rate based on the number of matching plans. More evaluation details are given in Appendix C.1.

## 5.2. Main Results

As shown in Table 1, we evaluate HTP's effectiveness across these benchmarks. We can get the following key results:

**1) HTP consistently achieves the best success rate across all tasks, regardless of the backbone model.** For example, the success rate of Gemini-1.5-Pro on TravelPlanner increased from 8.9% (EvoAgent) to 36.1% with HTP, representing a $4.06\times$ performance improvement. This highlights HTP's effectiveness, strong generalization across diverse benchmarks, and adaptability to different backbone models.

**2) HTP demonstrates more significant performance improvements on tasks with longer reasoning chains.** As shown in Appendix E.3, a single inference on the TravelPlanner and Mystery Blocksworld dataset typically requires over 60 reasoning steps, much longer than the approximately 30 steps needed for datasets like Blocksworld and Trip Planning. As a result, HTP achieves a $2.8\times$ and a $4.8\times$ performance improvement on the TravelPlanner and Mystery Blocksworld respectively, far surpassing the $1.3\times$ and $1.8\times$ improvements observed on the other two datasets based on GPT-4o. This improvement is attributed to the ability of HTP to flexibly implement a divide-and-conquer strategy, effectively reducing the length of reasoning chains while maintaining the integrity of the planning process.

## 5.3. Comparison with powerful SOTA LLMs

To comprehensively demonstrate the effectiveness of our method, we compare our approach with current SOTA closed-source models and prominent open-source models. As shown in Table 2, HTP outperforms the majority of powerful closed-source and open-source models with several

---

[2]Due to the unconventional experimental setups of MTP, we present the comparison results in the Appendix C.2.

*Table 2.* Comparison with leading LLMs across three tasks. The best results are **bolded**, and the second best ones are underlined. SR represents the Success Rate. The cost is calculated based on the API cost for a single instance.

| Benchmark | Model | SR | Cost(in $) |
|---|---|---|---|
| TravelPlanner | GPT-4o | 4.4 | 0.025 |
| | GPT-4 | 4.4 | 0.025 |
| | o1-preview | 10.0 | 0.380 |
| | o1-mini | 1.67 | 0.067 |
| | Gemini-1.5-Pro | 5.6 | 0.035 |
| | GPT-4o+HTP | 20.0 | 0.071 |
| | Gemini-1.5-Pro+HTP | **36.1** | 0.102 |
| Blocksworld | GPT-4o | 35.5 | 0.007 |
| | GPT-4 | 34.6 | 0.018 |
| | GPT-4-Turbo | 40.1 | 0.012 |
| | o1-preview | **97.8** | 0.420 |
| | o1-mini | 56.6 | 0.037 |
| | Gemini-1.5-Pro | 23.8 | 0.003 |
| | Claude 3.5(Sonnet) | 54.8 | 0.004 |
| | Claude 3(Opus) | 59.3 | 0.017 |
| | LLaMA-3.1-405B | 62.6 | - |
| | LLaMA-3.1-70B | 34.4 | - |
| | GPT-4o+HTP | 54.8 | 0.032 |
| | Gemini-1.5-Pro+HTP | 67.2 | 0.023 |
| Trip Planning | GPT-4o | 3.7 | 0.003 |
| | GPT-4 | 31.1 | 0.009 |
| | o1-preview | 36.2 | 0.530 |
| | Gemini-1.5-Pro | 34.8 | 0.005 |
| | LLaMA-3.1-70B | 32.5 | - |
| | GPT-4o+HTP | 36.9 | 0.046 |
| | Gemini-1.5-Pro+HTP | **42.8** | 0.069 |

hundred billions of parameters across three datasets. In addition, HTP significantly reduces token cost. Notably, HTP based on Gemini-1.5-Pro achieves 42.8% accuracy on the Natural Plan benchmark, exceeding the performance of o1-preview while requiring only 13% of its cost. Similarly, HTP based on GPT-4o attains performance equivalent to o1-preview, with the cost reduced to just 8.7%.

## 5.4. Ablation Study and Additional Analysis

**Ablations** To assess the impact of individual HTP modules on overall performance, we conduct an ablation study using GPT-4o and Gemini-1.5-Pro as the backbone models. Specifically, we evaluate five ablated versions of HTP, each omitting a distinct module. Detailed descriptions of these variants are provided in Appendix C.3. As shown in Table 3, removing any module consistently leads to a notable decline in performance. In particular:

**1) The hierarchical thinking mechanism plays a crucial role in enhancing LLM performance for planning problems.** The removal of the division module, which supports hierarchical thinking, results in a significant drop in success rates. This suggests that LLMs with hierarchical thinking capabilities achieve substantial improvements across various

*Table 3.* Ablation studies for key components of HTPAgent based on GPT-4o and Gemini-1.5-Pro, evaluating the success rate (SR) across TravelPlanner (Travel.), Blocksworld (Blocks.), and Trip Planning (Trip.).

| Model | Method | Travel. SR | Blocks. SR | Trip. SR |
|-------|--------|------------|------------|----------|
| GPT-4o | HTP | **20.0** | **54.7** | **36.9** |
| | w/o selection | 18.9 | 42.0 | 21.1 |
| | w/o division | 6.1 | 37.2 | 18.8 |
| | w/o decision | 17.8 | 48.5 | 15.3 |
| | w/o outline | 14.4 | 43.3 | 30.7 |
| | w/o self-guided | 8.3 | 42.2 | 36.6 |
| Gemini-1.5-Pro | HTP | **36.1** | **67.2** | **42.8** |
| | w/o selection | 34.4 | 55.8 | 36.6 |
| | w/o division | 7.2 | 27.8 | 38.4 |
| | w/o decision | 33.9 | 57.2 | 35.0 |
| | w/o outline | 30.6 | 56.0 | 37.2 |
| | w/o self-guided | 18.9 | 46.5 | 42.5 |

aspects of the solution process.

**2) The planning outline, in comparison to demonstration examples, significantly boosts the potential of LLMs.** For instance, GPT-4o-based HTP achieves a success rate of 54.7% on Blocksworld, whereas substituting the planning outline with a fixed hypertree-structured example reduces the success rate to 43.3%.

**3) The self-guided planning process markedly improves the performance of HTP on both the TravelPlanner and Blocksworld datasets.** For example, GPT-4o-based HTP achieves a success rate of 20.0 on the TravelPlanner dataset, which drops sharply to 8.3 when the self-guided planning process is removed. This highlights the importance of in-depth reasoning for solving subtasks. The self-guided planning process has a lesser effect on Trip Planning, primarily due to the simplicity of its subtasks.

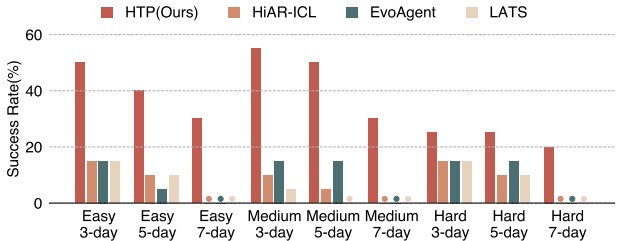

*Figure 4.* Success rates on the TravelPlanner benchmark categorized by problem instance difficulty and trip durations. Circles indicate instances with a success rate of 0 for clearer identification.

**Reasoning Difficulty** To evaluate the impact of planning difficulty on the success rates of different planning methods, we conducted a detailed analysis using the TravelPlanner benchmark. As outlined in Section 5.1, TravelPlanner can be categorized by trip durations. With every two-day increase in trip duration, there is a significant rise in the required

reasoning steps, constraints, and subtasks in the itinerary. Figure 4 shows the success rates for four methods across various trip durations based on Gemini-1.5-pro. The results reveal a noticeable decline for HiAR-ICL, Evoagent, and LATS as trip duration increases, while HTP maintains a more consistent performance trend. These findings highlight HTP's superior adaptability to more challenging planning tasks, demonstrating its robustness and scalability in scenarios with increasing planning complexity.

**Pruning Strategies** To determine the optimal hyperchain pruning strategy, we evaluate three selection methods: *(1) Width-based$_n$*: prune branches exceeding a maximum width of $n$. *(2) Probability-based$_n$*: retain only the top $n$ branches with the highest confidence probabilities. *(3) LLM-based$_n$*: use a LLM to select up to $n$ branches considered most promising. As shown in Table 3, both the selection and the decision module have a minimal impact on the TravelPlanner benchmark, which indicates that the TravelPlanner benchmark is less sensitive to pruning strategies. Therefore, we focus on the Blocksworld and Trip Planning datasets, with Gemini-1.5-Pro as the backbone model and GPT-4o employed for confidence scoring in the *Probability-based$_n$* method and decision-making in the *LLM-based$_n$* method.

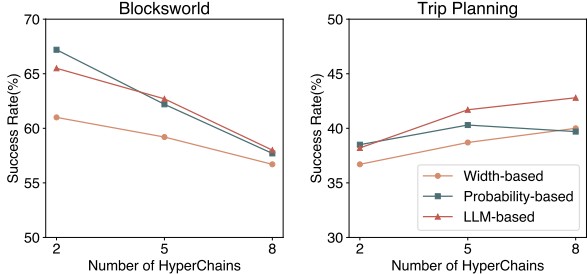

*Figure 5.* Success rates on the Blocksworld and Trip Planning benchmarks, categorized by different pruning strategies.

Figure 5 presents a detailed comparison of the strategies across the two tasks. Notably, in the Blocksworld task, peak performance is achieved at $n = 2$, with a slight decline as $n$ increases. In contrast, performance in the Trip Planning task improves steadily as $n$ increases, suggesting that Blocksworld is more likely to find an optimal path with fewer attempts, while Trip Planning benefits from exploring a larger number of branches. Additionally, when $n$ is small, the LLM-based and Probability-based strategies perform similarly, both outperforming the Width-based strategy. However, as $n$ increases, the LLM-based strategy outperforms the others, while the Probability-based approach shows a decline in performance. This discrepancy is likely due to biases in the confidence scores, which may accumulate inaccuracies at deeper levels of reasoning.

## 6. Limitations and Future Work

**Limitations**   While our approach consistently improves performance on complex planning tasks, several key limitations remain when compared to human planning. LLMs struggle with complex single-step reasoning, lack human prior knowledge (e.g., budgeting strategies), are vulnerable to long-horizon errors, and lack mechanisms for self-reflection and backtracking.

**Future Work**   Looking ahead, our work offers several promising directions for future research. First, HTP integrates naturally with self-reflection and backtracking, making it well-suited for real-world tasks such as meeting planning and calendar scheduling. Its hierarchical hyperchain structure enables more accurate error correction by avoiding redundant reasoning over unrelated paths. Second, HTP's scalability and adaptability make it a strong candidate for autonomous agent decision-making. Future work may explore equipping end-to-end agents with hierarchical reasoning abilities using HTP. Third, combining HTP with LLM-based heuristic reward functions presents a promising path for improving decision quality and learning efficiency.

## 7. Conclusion

In this work, we introduce HTP, a novel planning paradigm that utilizes hypertree construction for hierarchical thinking. By iteratively expanding the hypertree through a top-down process, HTP enables a multi-level divide-and-conquer strategy to create an effective planning outline. In our evaluation across three benchmarks, HTP consistently outperforms all models. Its ability to transform extensive inference chains into hypertree structures led to significant performance improvements, especially on datasets with long inference chains. These results highlight the effectiveness of HTP in tackling complex planning challenges.

## Acknowledgements

This work was supported in part by National Key R&D Program of China under contract 2022ZD0119801, National Nature Science Foundations of China grants U23A20388, 62021001. This work was supported in part by Huawei as well. We would like to thank all the anonymous reviewers for their insightful comments.

## Impact Statement

The HTP paradigm marks a major breakthrough in the reasoning field, especially for complex planning tasks. As the first method to model the reasoning process with a hypertree structure, it enables hierarchical thinking in LLMs, addressing key reasoning challenges and providing a solid structural basis for future research. The top-down hypertree construction algorithm offers LLMs structural guidance for solving complex problems, enhancing decision-making accuracy and efficiency.

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

# A. More Background

**Reasoning path modeling** LLMs are increasingly used for inference, with reasoning path modeling methods evolving beyond traditional CoT and ToT to decompose complex problems into simpler tasks. Program of Thought (PoT) (Chen et al., 2023b) structures reasoning as programmatic steps, using variable names for semantic clarity. Algorithm of Thought (AoT) (Sel et al., 2024) compacts these steps into a single prompt, reducing token consumption while improving efficiency. Graph of Thought (GoT) (Besta et al., 2024) models inference as a graph, enabling dynamic path selection and backtracking. Forest of Thought (FoT) (Bi et al., 2024) builds an ensemble of inference trees, leveraging sparse activation for more efficient decision-making. These methods enhance LLMs' ability to handle complex problems with greater flexibility and accuracy. Meanwhile, complex reasoning has also been widely explored in circuit design (Wang et al., 2024e; 2025), logic synthesis (Bai et al., 2025; Wang et al., 2024d;a), and optimization (Wang et al., 2023b), demonstrating its potential in guiding structured and efficient decision-making. Notably, hierarchical and sample-efficient frameworks (Wang et al., 2022; 2023c) have shown success in these domains, motivating their extension to hierarchical reasoning with LLMs.

# B. Experiment Settings

## B.1. Implementation Details of the Baselines

Below, we provide short descriptions of the five planning strategy baseline methods and two agent methods.

- **CoT:** CoT (Chain-of-Thought) (Kojima et al., 2022) facilitates efficient and effective step-by-step reasoning by appending the phrase "Let's think step by step" to the prompt, guiding the model through a structured reasoning process.

- **ToT:** ToT (Tree-of-Thought) (Yao et al., 2024) enables multi-path reasoning by guiding large models to generate multiple feasible parallel reasoning paths simultaneously. In our setup, the pruning strategy of ToT is always aligned with the HTP method.

- **LATS:** LATS refers to Language Agent Tree Search (Zhou et al., 2023), a general framework that harnesses the reasoning, acting, and planning capabilities of language models while integrating Monte Carlo Tree Search to explore the search space. It has demonstrated exceptional performance in planning tasks (Chen et al., 2024).

- **One shot:** The one-shot approach enhances the reasoning ability of large models by providing a fixed example, guiding them to learn through analogy. For specific examples, please refer to Appendix E.4.

- **HiAR-ICL:** HiAR-ICL stands for High-level Automated Reasoning Paradigm in In-Context Learning (Wu et al., 2024), a powerful ICL method that shifts the focus from specific examples to abstract thinking patterns, thereby enhancing the generalization ability of LLMs.

- **RAP:** RAP (Reasoning via Planning) (Hao et al., 2023) repurposes the LLM as both a world model and a reasoning agent, integrating a principled planning algorithm based on Monte Carlo Tree Search to enable strategic exploration within the vast reasoning space.

- **EvoAgent:** EVOAGENT (Yuan et al., 2024) is a generic method to automatically extend expert agents to multi-agent systems via the evolutionary algorithm, thereby improving the effectiveness of LLM-based agents in solving tasks.

- **MTP**: MTP(meta-task planner) (Zhang et al., 2024a) is a zero-shot methodology for collaborative LLM-based multi-agent systems that simplifies complex task planning by decomposing it into a hierarchy of subordinate tasks.

## B.2. Backbone Model Selection

For OpenAI models, we use gpt-3.5-turbo-1106 and gpt-4o-2024-08-06. For Gemini-1.5-Pro, we use Google Gemini-1.5-Pro APIs to obtain results. We set the temperature to 0 for all models.

# C. Additional Results

## C.1. Evaluation Details of TravelPlanner

Grounding to travel planning, a real-world use-case that inherently involves various constraints like user preferences and commonsense rules, TravelPlanner evaluates whether agents can formulate flexible travel plans using gathered information to meet these constraints. We test EVOAGENT and all baselines on the TravelPlanner validation set, which consists of 180 user queries with the collected information. To evaluate the travel plans generated by agents, TravelPlanner adopts the following evaluation metrics:

1. Delivery Rate: Assesses if agents can complete a plan within a limited number of steps (30 in our experimental setting). Failures are due to dead loops, numerous failed attempts, or exceeding the step limit.

2. Commonsense Constraint Pass Rate: Evaluates if an agent can incorporate commonsense into their plan.

3. Hard Constraint Pass Rate: Measures if a plan meets all explicit hard constraints in the query, testing the agent's ability to adapt to diverse user preferences.

4. Success rate: Indicates the proportion of viable plans that meet all criteria, reflecting the agent's proficiency in creating practical plans.

Furthermore, TravelPlanner uses micro and macro strategies to assess the Commonsense and Hard Constraint Pass Rates. The micro strategy calculates the ratio of met constraints to the total. The macro strategy measures the proportion of plans that meet all commonsense or hard constraints. Together, these strategies assess an agent's ability to satisfy individual constraints and all constraints comprehensively.

## C.2. Comparision with MTP

In this part, we compare our HyperTree Planning (HTP) method with the Meta-task Planner (MTP) (Zhang et al., 2024a) approach. As MTP reports its results exclusively on 3-day trip durations within the TravelPlanner benchmarks, we adopt the same evaluation setting to ensure consistency. Furthermore, we introduce three additional baseline methods for comparison: React (Yao et al., 2022), Direct, and CoT (Wei et al., 2022). Importantly, both the React and MTP methods employ a two-stage mode, while the Direct, CoT, and HTP methods utilize a sole-planning mode, bypassing the information-gathering step for a more streamlined process.

From Table 4, 5, and 6, it is clear that HTP consistently achieves the highest success rate across tasks of varying difficulty levels: easy, medium, and hard. This superiority is even more evident in the detailed metrics of Commonsense and Hard Constraint, underscoring the effectiveness of our approach in comprehensively addressing complex problems.

*Table 4.* The Success Rates (%) on **Easy** Instances. The highest success rates are highlighted in bold blue.

| Methods | | | GPT-4 + React | GPT-4 + MTP | GPT-4 + Direct | GPT-4 + CoT | GPT-4 + HTP(Ours) |
|---|---|---|---|---|---|---|---|
| | Delivery Rate | | 95.00 | 100.00 | 100.00 | 100.00 | 100.00 |
| | Common- | Micro | 75.00 | 95.63 | 95.00 | 89.38 | 95.63 |
| Validation | sense | Macro | 5.00 | 70.00 | 65.00 | 55.00 | 65.00 |
| Set | Hard | Micro | 15.00 | 55.00 | 60.00 | 55.00 | 60.00 |
| | Constraint | Macro | 15.00 | 55.00 | 60.00 | 55.00 | 60.00 |
| | Success Rate | | 5.00 | **55.00** | 35.00 | 40.00 | **55.00** |

*Table 5.* The Success Rates (%) on **Medium** Instances. The highest success rates are highlighted in bold blue.

| Methods | | | GPT-4 + React | GPT-4 + MTP | GPT-4 + Direct | GPT-4 + CoT | GPT-4 + HTP(Ours) |
|---|---|---|---|---|---|---|---|
| | Delivery Rate | | 100.00 | 95.00 | 100.00 | 100.00 | 100.00 |
| | Common- | Micro | 83.75 | 82.50 | 91.88 | 84.38 | 96.88 |
| Validation | sense | Macro | 10.00 | 20.00 | 50.00 | 55.00 | 75.00 |
| Set | Hard | Micro | 7.50 | 55.00 | 55.00 | 55.00 | 80.00 |
| | Constraint | Macro | 0.00 | 55.00 | 20.00 | 25.00 | 70.00 |
| | Success Rate | | 0.00 | 15.00 | 5.00 | 10.00 | **55.00** |

*Table 6.* The Success Rates (%) on **Hard** Instances. The highest success rates are highlighted in bold blue.

| Methods | | | GPT-4 + React | GPT-4 + MTP | GPT-4 + Direct | GPT-4 + CoT | GPT-4 + HTP(Ours) |
|---|---|---|---|---|---|---|---|
| | Delivery Rate | | 100.00 | 95.00 | 100.00 | 100.00 | 100.00 |
| | Common- | Micro | 79.38 | 83.75 | 89.38 | 88.75 | 94.38 |
| Validation | sense | Macro | 10.00 | 40.00 | 35.00 | 45.00 | 60.00 |
| Set | Hard | Micro | 5.00 | 41.25 | 50.00 | 55.00 | 75.00 |
| | Constraint | Macro | 0.00 | 30.00 | 5.00 | 5.00 | 45.00 |
| | Success Rate | | 0.00 | 30.00 | 0.00 | 0.00 | **35.00** |

## C.3. Details of Ablations

We evaluate five ablated versions of HTP using GPT-4 as the backbone model, where each version omits one specific design principle:

1)*w/o selection:* The selection module is replaced with a strategy that always chooses the **leftmost** leaf node.

2)*w/o division:* The decomposition module is removed. The hypertree-structure planning outline degenerates into tree-structure.

3)*w/o decision:* The decision module is removed, and during expansion, a single branch is always maintained for each non-leaf node.

4)*w/o outline:* The planning outline is replaced with a fixed hypertree planning example instead of being dynamically generated for each query.

5)*w/o self-guided:* The hypertree results are directly used as the complete planning content, but all necessary external information is provided during the construction of the hypertree.

## D. Analysis of Computational Cost

To address concerns regarding the efficiency of our proposed method, we provide a comprehensive analysis of computational cost across different reasoning paradigms, including CoT (Wei et al., 2022), RAP (Hao et al., 2023), and our method HTP, using the **TravelPlanner** dataset. Due to the suboptimal performance of open-source models on this task, we adopt GPT-4o as the backbone model for all evaluations.

We assess computational efficiency from three perspectives: inference speed, token cost, and computational complexity. Let $n$ be the number of branches expanded at each step (with $n \leq 2$ in TravelPlanner), $l$ the average number of reasoning steps per chain, and $k$ the number of sampled trajectories in MCTS-based methods.

| Model | Inference Speed (s) | Token Cost (in/out) | Computational Complexity |
|---|---|---|---|
| CoT | 6.92 | 4328 / 641 | $\mathcal{O}(l)$ |
| RAP | 41.94 | 5440 / 3374 | $\mathcal{O}(nkl)$ |
| **HTP** | **25.27** | **5562 / 963** | $\mathcal{O}(nl)$ |

*Table 7.* Comparison of computational cost across different reasoning methods on TravelPlanner.

The results clearly demonstrate that HTP significantly reduces computational cost compared to RAP across all three metrics. This is attributed to the elimination of MCTS in HTP, which avoids costly sampling and value estimation procedures, thereby improving efficiency.

Additionally, although HTP employs a hyperchain reasoning structure to support hierarchical planning, this does not increase its overall computational complexity. The higher dimensionality of reasoning is counterbalanced by shorter average reasoning paths, resulting in computational efficiency that remains on par with non-hierarchical approaches.

**Additional Analysis and Potential Improvements.** We note that the inference cost of HTP can sometimes exceed that of conventional single-path backbone models, primarily due to two factors:

1. **Multi-path reasoning overhead**, resulting from the expansion of multiple candidate paths (e.g., tree search).

2. **Hierarchical decomposition**, where the original task is divided into finer-grained subproblems, requiring more detailed reasoning.

The impact of these factors varies across tasks. In the TravelPlanner task, the overhead mainly stems from the need for constraint-aware reasoning (factor 2), whereas in NaturalPlan, the major cost is associated with the search process needed to identify a feasible solution via trial-and-error (factor 1).

Despite these additional sources of cost, HTP remains substantially more efficient than MCTS-based methods, as it avoids repeated simulations and value estimation.

To further improve efficiency, we propose two directions for future work:

- **Using smaller fine-tuned models** (instead of large backbone models) for intermediate reasoning steps during hypertree construction.

- **Adopting meta-learning techniques** to predict task complexity and dynamically adjust hypertree parameters (e.g., depth and width), allowing for adaptive and efficient reasoning based on task demands.

## E. Case Study

This includes examples of different states on the four task sets of TravelPlanner, Blocksworld, Mystery Blocksworld, and Trip Planning. E.1 include examples of the HyperTree library (rules and nodes). E.2 include examples of the HyperTree-stuctured planning outline. E.3 include examples of the planning process. E.4 include examples of the final plan.

### E.1. Examples of the HyperTree Library (Rules and Nodes)

**TravelPlanner:**

```
Rules:                                                                    1
1. [Plan] -> [Transportation][Accommodation][Attraction][Dining] # The plan can be divided 2
 into four aspects.
2. [Transportation] -> {{Specific segments of transportation}} # Break transportation down 3
 into specific transportation choices for each segment of the trip.
3. [transportation from A to B] -> [Self-driving][Taxi][Flight] # The transportation mode 4
for each segment can be choose from self-driving, taxi, and flights.
4. [Self-driving] -> [transportation availability][transportation preference][cost][non-    5
conflicting] # For each mode of transportation, you should consider transportation
availability, transportation preference, cost and non-conflicting.
   [Taxi] -> [transportation availability][transportation preference][cost][non-            6
conflicting]
   [Flight] -> [transportation availability][transportation preference][cost][non-          7
conflicting]
5. [Accommodation] -> {{Specific accommodation for each city}} # Break accommodation down   8
into specific accommodation options for each city.
6. [Accommodation for A] -> [cost][house rule][room type][minimum stay] # For              9
accommodation in each city, consider the cost, house rules, room type, and minimum stay
requirements.
7. [Attraction] -> {{specific attraction for each city}} # Break accommodation down into   10
specific attraction options for each city.
8. [Dining] -> {{Specific dining for each city}} # Break dining down into specific dining   11
options for each city.
9. [Dining for A] -> [cost][cuisine] # For dining in each city, consider the cost and      12
cuisine.
                                                                          13
Divisible Nodes:                                                          14
[Plan]; [Transportation]; [Taxi]; [Self-driving]; [Flight]; [Accommodation]; [Attraction]; 15
 [Dining];
{{Specific segment of transportation}} # (Note: This placeholder represents the specific   16
modes of transportation for one segment of the trip, such as [transportation from A to B])
;
{{Specific accommodation for one city}} # (Note: This placeholder represents the specific   17
accommodation options for one city in the trip, such as [Accommodation for A]);
```

```
{{Specific dining for one city}} # (Note: This placeholder represents the specific dining    18
options for one city in the trip, such as [Dining for A]);
                                                                                             19
Leaf Nodes(Example):                                                                         20
[transportation availability]; [transportation preference]; [transportation cost]; [house    21
rule]; [room type]; [accommodation cost]; [minimum stay]; [cuisine]; [dining cost];
{{specific attraction for one city}} # (Note: This placeholder represents the specific       22
attraction options for one city in the trip, such as [attraction for A]).
```

## Blocksworld:

```
Rules:                                                                                        1
[Plan] -> {{[[{{Block}} on the table][{{Block}} on the table][{{Block}} on top of {{Block     2
}}]] [{{Block}} on top of {{Block}}]]}} # (Note: The quantity and types of symbols are
indefinite. The plan is divided into different floors, and if the target state requires
multiple blocks on the same floor, there will result in multiple symbols in parallel
relationship);
[{{Block}} on the table] -> {{[to get {{Block}} clear][to get hand empty][to get {{Block}}    3
 on the table]}} # (Note: The quantity and types of symbols are indefinite. The general
approach is to break down the task into atomized, specific actions, and there can be
multiple symbols in parallel relationships);
[{{Block}} on top of {{Block}}]  -> {{[to get {{Block}} clear][to get hand empty][to get      4
{{Block}} clear][to get hand empty][to get {{Block}} on top of {{Block}}]}} # (Note: The
quantity and types of symbols are indefinite. The general approach is to break down the
task into atomized, specific actions, and there can be multiple symbols in parallel
relationships);
                                                                                              5
Divisible Nodes:                                                                              6
[Plan];                                                                                       7
[{{Block}} on the table] # (Note: This placeholder represents the specific blocks);           8
[{{Block}} on top of {{Block}}] # (Note: This placeholder represents the desired              9
arrangement of the blocks);
                                                                                             10
Leaf Nodes(Example):                                                                         11
[to get {{Block}} clear] # (Note: This placeholder represents the specific blocks);          12
[to get hand empty];                                                                         13
[to get {{Block}} on the table]; [to get {{Block}} on top of {{Block}}] # (Note: Unlike      14
non-terminals which refer to broad directions, here it refers to atomized specific tasks.)
;
```

## Mystery Blocksworld:

```
Rules:                                                                                        1
[Plan] -> {{[[Planet {{Object}}][Planet {{Object}}][{{Object}} Craves {{Object}}] [{{Object   2
}} Craves {{Object}}]]}} # (Note: The quantity and types of symbols are indefinite. The
plan is divided into different Crave-relationship, and if the target state requires
multiple objects on similar Crave-relationship, there will result in multiple symbols in
parallel relationships);
[Planet {{Object}}] -> {{[to get Province {{Object}} becomes True][to get Harmony becomes      3
True][to get Planet {{Object}} becomes True]}} # (Note: The quantity and types of symbols
are indefinite. The general approach is to break down the task into atomized, specific
actions, and there can be multiple symbols in parallel relationships);
[{{Object}} Craves {{Object}}] -> {{[to get Province {{Object}} becomes True][to get          4
Harmony becomes True][to get Province {{Object}} becomes True][to get Harmony becomes True
][to get {{Object}} craves {{Object}}]}} # (Note: The quantity and types of symbols are
indefinite. The general approach is to break down the task into atomized, specific actions
, and there can be multiple symbols in parallel relationships);
                                                                                              5
Divisible Nodes:                                                                              6
[Plan];                                                                                       7
[Planet {{Object}}] # (Note: This placeholder represents specific objects on a planet,        8
such as [Planet object A]);
[{{Object}} Craves {{Object}}] # (Note: This placeholder represents the desired               9
arrangement where one object craves another);
                                                                                             10
```

```
Leaf Nodes(Example):                                                              11
[to get Province {{Object}} becomes True]# (Note: This placeholder represents specific   12
objects that need to be cleared);
[to get Harmony becomes True]; [to get Planet {{Object}} becomes True]; [to get {{Object}}   13
 craves {{Object}}] # (Note: Unlike non-terminals which refer to broad directions, here it
 refers to atomized specific tasks.);
```

## Trip Planning:

```
Rules:                                                                            1
[Plan] -> [Cities with determine dates][Cities with undetermine dates] # (Note: Translate   2
the travel plan into cities with specific dates and cities with unspecific dates, and
consider them separately);
[Cities with determine dates] -> {{[{{City}}][{{City}}]}}  # (Note: The quantity of   3
symbols are indefinite. The cities here are those where the activities are explicitly
specified in the query);
[Cities with undetermine dates] -> {{[{{City}}][{{City}}]}} # (Note: The quantity and   4
types of symbols are indefinite. The cities here are the remaining cities in the query);
[{{City}}] -> [from day {{i}} to day {j}] # (Note: Further expand the city into specific   5
dates).
                                                                                  6
Devisible Nodes:                                                                  7
[Plan]; [Cities with determine dates]; [Cities with undetermine dates];           8
[{{City}}] # (Note: This placeholder represents the city in the query, such as [London]);   9
                                                                                  10
Leaf Nodes(Example):                                                              11
[from day {{i}} to day {j}] # (Note: This placeholder represents specific date in one city   12
);
```

## E.2. Examples of the HyperTree-stuctured Planning Outline

## TravelPlanner:

```
[Plan]                                                                            1
    [Transportation]                                                              2
        [Transportation from Fort Lauderdale to City 1 in Georgia]                3
            [Self-driving]                                                        4
                [transportation availability]                                     5
                [transportation preference]                                       6
                [transportation cost]                                             7
            [Taxi]                                                                8
                [transportation availability]                                     9
                [transportation preference]                                       10
                [transportation cost]                                             11
            [Flight]                                                              12
                [transportation availability]                                     13
                [transportation preference]                                       14
                [transportation cost]                                             15
        [Transportation from City 1 in Georgia to City 2 in Georgia]              16
            [Self-driving]                                                        17
                [transportation availability]                                     18
                [transportation preference]                                       19
                [transportation cost]                                             20
            [Taxi]                                                                21
                [transportation availability]                                     22
                [transportation preference]                                       23
                [transportation cost]                                             24
            [Flight]                                                              25
                [transportation availability]                                     26
                [transportation preference]                                       27
                [transportation cost]                                             28
        [Transportation from City 2 in Georgia to City 3 in Georgia]              29
            [Self-driving]                                                        30
                [transportation availability]                                     31
```

```
                    [transportation preference]                                    32
                    [transportation cost]                                          33
                [Taxi]                                                             34
                    [transportation availability]                                  35
                    [transportation preference]                                    36
                    [transportation cost]                                          37
                [Flight]                                                           38
                    [transportation availability]                                  39
                    [transportation preference]                                    40
                    [transportation cost]                                          41
            [Transportation from City 3 in Georgia to Fort Lauderdale]            42
                [Self-driving]                                                     43
                    [transportation availability]                                  44
                    [transportation preference]                                    45
                    [transportation cost]                                          46
                [Taxi]                                                             47
                    [transportation availability]                                  48
                    [transportation preference]                                    49
                    [transportation cost]                                          50
                [Flight]                                                           51
                    [transportation availability]                                  52
                    [transportation preference]                                    53
                    [transportation cost]                                          54
    [Accommodation]                                                               55
        [Accommodation for City 1 in Georgia]                                     56
            [minimum stay]                                                        57
            [house rule]                                                          58
            [room type]                                                           59
            [accommodation cost]                                                  60
        [Accommodation for City 2 in Georgia]                                     61
            [minimum stay]                                                        62
            [house rule]                                                          63
            [room type]                                                           64
            [accommodation cost]                                                  65
        [Accommodation for City 3 in Georgia]                                     66
            [minimum stay]                                                        67
            [house rule]                                                          68
            [room type]                                                           69
            [accommodation cost]                                                  70
    [Attraction]                                                                  71
        [Attraction for City 1 in Georgia]                                        72
        [Attraction for City 2 in Georgia]                                        73
        [Attraction for City 3 in Georgia]                                        74
    [Dining]                                                                      75
        [Dining for City 1 in Georgia]                                           76
            [cuisine]                                                             77
            [dining cost]                                                         78
        [Dining for City 2 in Georgia]                                           79
            [cuisine]                                                             80
            [dining cost]                                                         81
        [Dining for City 3 in Georgia]                                           82
            [cuisine]                                                             83
            [dining cost]                                                         84
```

## Blocksworld:

```
[Plan]                                                                            1
    [Blue block on the table]                                                     2
        [to get the blue block clear]                                             3
        [to get the blue block on the table]                                      4
    [Orange block on the table]                                                   5
        [to get the orange block clear]                                           6
        [to get the orange block on the table]                                    7
    [Orange block on top of Blue block]                                           8
        [to get the blue block clear]                                             9
```

```
            [to get the orange block clear]                                     10
            [to get the orange block on top of the blue block]                  11
        [Red block on top of Orange block]                                      12
            [to get the red block clear]                                        13
            [to get the orange block clear]                                     14
            [to get the red block on top of the orange block]                   15
```

**Mystery Blocksworld:**

```
[Plan]                                                                          1
    [Planet object b]                                                           2
        [to get Province object b becomes True]                                 3
        [to get Planet object b becomes True]                                   4
    [Object d Craves Object b]                                                  5
        [to get Province object d becomes True]                                 6
        [to get Province object b becomes True]                                 7
        [to get object d Crave object b]                                        8
    [Object c Craves Object d]                                                  9
        [to get Province object c becomes True]                                 10
        [to get Province object d becomes True]                                 11
        [to get object c Crave object d]                                        12
```

**Trip Planning:**

```
[Plan]                                                                          1
    [Cities with determine dates]                                               2
        [Valencia]                                                              3
            [from day 1 to day 3]                                               4
        [Stockholm]                                                             5
            [from day 6 to day 10]                                              6
        [Madrid]                                                                7
            [from day 20 to day 21]                                             8
    [Cities with undetermine dates]                                             9
        [Seville]                                                               10
            [from day 3 to day 4]                                               11
        [Bucharest]                                                             12
            [from day 21 to day 22]                                             13
        [Nice]                                                                  14
            [from day 17 to day 20]                                             15
        [Manchester]                                                            16
            [from day 4 to day 6]                                               17
        [Krakow]                                                                18
            [from day 10 to day 11]                                             19
        [Vilnius]                                                               20
            [from day 11 to day 14]                                             21
        [Zurich]                                                                22
            [from day 14 to day 17]                                             23
```

## E.3. Examples of the Planning Process

**TravelPlanner:**

```
Break down the travel planning into four parts: transportation, attractions, dining, and  1
accommodation.
        For the transportation part:                                            2
        The transportation I need to consider involves four parts: from Houston to  3
Nashville on 2022-03-21, from Nashville to Knoxville on 2022-03-23, from Knoxville to
Chattanooga on 2022-03-25, from Chattanooga to Houston on 2022-03-27.
                For the transportation from Houston to Nashville:               4
                I observe that flights, self-driving, and taxis are all available.  5
                The user prefer no self-driving, so I can only choose between Flight or  6
taxi for transportation. I should choose the one that costs less.
                Now calculate the cost of choosing the flight option: the lowest-priced  7
flight is $145, and there are 2 travelers, making the total cost $145 * 2 = $290.
```

Now calculate the cost of choosing the taxi option: there are 2 travelers, 8
we need 1 taxi, the price is $1253 * 1 = $1253.

So I will choose the flight option. I will submit: "Flight Number: 9
F3956409, from Houston to Nashville, Departure Time: 17:36, Arrival Time: 19:12".

For the transportation from Nashville to Knoxville on 2022-03-23: 10

I observed that there is no flight from Nashville to Knoxville on 11
2022-03-23, but self-driving and taxis are available.

The user prefer no self-driving, so I can only choose taxi. I will submit: 12
"taxi, from Nashville to Knoxville, duration: 2 hours 42 mins".

For the transportation from Knoxville to Chattanooga on 2022-03-25: 13

I observed that there is no flight from Knoxville to Chattanooga on 14
2022-03-25, but self-driving and taxis are available.

The user prefer no self-driving, so I can only choose taxi. I will submit: 15
"taxi, from Knoxville to Chattanooga, duration: 1 hour 41 mins".

For the transportation from Chattanooga to Houston on 2022-03-27: 16

I observed that there is no flight Chattanooga to Houston on 2022-03-27, 17
but self-driving and taxis are available.

The user prefer no self-driving, so I can only choose taxi. I will submit: 18
"taxi, from Chattanooga to Houston, duration: 11 hours 47 mins".

For the attractions part: 19

The attractions I need to consider involves three parts: attractions in Nashville, 20
Knoxville, Chattanooga.

For the attractions in Nashville: 21

I am in Nashville from 2022-03-21 to 2022-03-22, so I should choose two 22
attractions in Nashville. I will submit: "Country Music Hall of Fame and Museum, Nashville
" for day 1 and "Nashville Zoo at Grassmere, Nashville" for day 2.

For the attractions in Knoxville: 23

I am in Knoxville from 2022-03-23 to 2022-03-24, so I should choose two 24
attractions in Knoxville. I will submit: "World's Fair Park, Knoxville" for day 3 and "
Knoxville Museum of Art, Knoxville" for day 4.

For the attractions in Chattanooga: 25

I am in Chattanooga from 2022-03-25 to 2022-03-27, so I have chosen three 26
attractions in Chattanooga. I will submit: "The Chattanooga Zoo at Warner Park,
Chattanooga" for day 5, "Rock City Gardens, Chattanooga" for day 6 and "Tennessee Aquarium
, Chattanooga" for day 7.

For the dining part: 27

The dining I need to consider involves three parts: dining in Nashville from 28
2022-03-21 to 2022-03-22, in Knoxville from 2022-03-23 to 2022-03-24, in Chattanooga from
2022-03-25 to 2022-03-27.

For the dining in Nashville from 2022-03-21 to 2022-03-22: 29

Since breakfast and lunch on the first day do not need to be considered, I 30
need to account for 4 meals.

The user requests French cuisine, and the French restaurant with the 31
lowest price is: Twigly.

The user requests Mexican cuisine, and the Mexican restaurant with the 32
lowest price is: Bablu Fast Food.

I still need to find 4 - 2 = 2 more restaurants, and the 2 with the lowest 33
prices are: Kitchen King, Govinda's Confectionery.

To sum up, I will submit: "Twigly, Nashville" for day 1 and "Bablu Fast 34
Food, Nashville", "Kitchen King, Nashville", "Govinda's Confectionery, Nashville" for day
2.

For the dining in Knoxville from 2022-03-23 to 2022-03-24: 35

I need to select 6 different restaurants. 36

The user requests French cuisine, and the French restaurant with the 37
lowest price is: Biryani By Kilo.

The user requests Mexican cuisine, and the Mexican restaurant with the 38
lowest price is: Open Kitchen.

I still need to find 6 - 2 = 4 more restaurants, and the 4 with the lowest 39
prices are: Chit Chat, Mamagoto, La-Nawaab, Tandoori Tadka.

To sum up, I will submit: "Biryani By Kilo, Knoxville", "Open Kitchen, 40
Knoxville", "Chit Chat, Knoxville" for day 3 and "Mamagoto, Knoxville", "La-Nawaab,
Knoxville", "Tandoori Tadka, Knoxville" for day 4.

For the dining in Chattanooga from 2022-03-25 to 2022-03-27: 41

Since lunch and dinner on the last day do not need to be considered, I 42
need to select 9 - 2 = 7 different restaurants.

The user requests French cuisine, and the French restaurant with the lowest price is: Tpot. [43]

The user requests Mexican cuisine, and the Mexican restaurant with the lowest price is: Liquid. [44]

I still need to find 7 − 2 = 5 more restaurants, and the 5 with the lowest prices are: Muradabadi, Burger's King, Basil Tree, Sardar A Pure Meat Shop, Pizza Hut Delivery. [45]

To sum up, I will submit: "Tpot, Chattanooga", "Liquid, Chattanooga", "Muradabadi, Chattanooga" for day 5, "Burger's King, Chattanooga", "Basil Tree, Chattanooga", "Sardar A Pure Meat Shop, Chattanooga" for day 6, and "Pizza Hut Delivery, Chattanooga" for day 7. [46]

For the accommodation part: [47]

The accommodation I need to consider involves three parts: accommodation in Nashville from 2022-03-21 to 2022-03-22, in Knoxville from 2022-03-23 to 2022-03-24, in Chattanooga from 2022-03-25 to 2022-03-27. [48]

For the accommodation in Nashville from 2022-03-21 to 2022-03-22: [49]

The user requests private room, the accommodations filtered based on the criterion are: Clean and large bedroom in a private house, Lovely room in heart of Williamsburg, FiDi Cozy room overlooking East River, Cozy bedroom close to Manhattan. [50]

The user requests house that allow smoking, the accommodations further filtered based on the criterion are: Lovely room in heart of Williamsburg, FiDi Cozy room overlooking East River, Cozy bedroom close to Manhattan. [51]

Since the stay is only for two nights, the minimum nights for accommodation should be limited to less than 3, and the accommodations further filtered based on the criterion are: Lovely room in heart of Williamsburg, FiDi Cozy room overlooking East River, Cozy bedroom close to Manhattan. [52]

Among the filtered accommodations, I should choose the one with the lowest price: [53]

Lovely room in heart of Williamsburg has a maximum occupancy of 4 people and is priced at $61. Since we are 2 people, we need 1 room, the total price is $61 * 1 = $61. [54]

FiDi Cozy room overlooking East River has a maximum occupancy of 5 people and is priced at $870. Since we are 2 people, we need 1 room, the total price is $870 * 1 = $870. [55]

Cozy bedroom close to Manhattan has a maximum occupancy of 3 people and is priced at $576. Since we are 2 people, we need 1 room, the total price is $576 * 1 = $576. [56]

Based on the calculations above, I will submit: "Lovely room in heart of Williamsburg, Nashville". [57]

For the accommodation in Knoxville from 2022-03-23 to 2022-03-24: [58]

The user requests private room, the accommodations that meet the criteria are: Cozy Private Room in Chinatown/ Lower East Side, The Diamond Room, Light-filled Room in Renovated Apt, Private Room, Beautiful & Private Manhattan Room, Brooklyn Sunny room 5 min to subway, Private 1 Bdrm Suite in Historic Brownstone, Charming bedroom with huge terrace in Greenpoint, Private large room near LGA airport with queen bed. [59]

The user requests house that allow smoking, the accommodations further filtered based on this criterion are: Cozy Private Room in Chinatown/ Lower East Side, The Diamond Room, Light-filled Room in Renovated Apt, Private Room, Brooklyn Sunny room 5 min to subway, Private 1 Bdrm Suite in Historic Brownstone, Charming bedroom with huge terrace in Greenpoint, Private large room near LGA airport with queen bed. [60]

Since the stay is only for two nights, the minimum nights for accommodation should be limited to less than 3, and the accommodations further filtered based on the criterion are: The Diamond Room, Light-filled Room in Renovated Apt, Private Room, Brooklyn Sunny room 5 min to subway, Private 1 Bdrm Suite in Historic Brownstone, Charming bedroom with huge terrace in Greenpoint, Private large room near LGA airport with queen bed. [61]

Among the filtered accommodations, I should choose the one with the lowest price: [62]

The Diamond Room has a maximum occupancy of 1 people and is priced at $1008. Since we are 2 people, we need 2 room, the total price is $1008 * 2 = $2016. [63]

Light-filled Room in Renovated Apt has a maximum occupancy of 2 people and is priced at $310. Since we are 2 people, we need 1 room, the total price is $310 * 1 = $310. [64]

Private Room has a maximum occupancy of 1 people and is priced at $922. Since we are 2 people, we need 2 room, the total price is $922 * 2 = $1844. [65]

Brooklyn Sunny room 5 min to subway has a maximum occupancy of 2 | 66
people and is priced at $793. Since we are 2 people, we need 1 room, the total price is
$793 * 1 = $793.

Private 1 Bdrm Suite in Historic Brownstone has a maximum | 67
occupancy of 2 people and is priced at $479. Since we are 2 people, we need 1 room, the
total price is $479 * 1 = $479.

Charming bedroom with huge terrace in Greenpoint has a maximum | 68
occupancy of 1 people and is priced at $712. Since we are 2 people, we need 2 room, the
total price is $712 * 2 = $1424.

Private large room near LGA airport with queen bed has a maximum | 69
occupancy of 1 people and is priced at $552. Since we are 2 people, we need 2 room, the
total price is $552 * 2 = $1104.

Based on the calculations above, I will submit: "Light-filled Room | 70
 in Renovated Apt, Knoxville".

For the accommodation in Chattanooga from 2022-03-25 to 2022-03-27: | 71

The user requests private room, the accommodations filtered based on the | 72
criterion are: Affordable Private Spacious Room in Brooklyn, Sunny room+Pvte office in
huge loft, Extra Cozy Room in Center of Williamsburg, Fort Greene Room, Cozy room in
Bushwick– 15 min to the city, Modern apartment w/ gorgeous view, Artsy Private BR in Fort
Greene Cumberland, Studio Deluxe 1 – Wyndham Midtown 45.

The user requests house that allow smoking, the accommodations further | 73
filtered based on the criterion are: Affordable Private Spacious Room in Brooklyn, Sunny
room+Pvte office in huge loft, Extra Cozy Room in Center of Williamsburg, Fort Greene Room
, Cozy room in Bushwick– 15 min to the city, Modern apartment w/ gorgeous view, Artsy
Private BR in Fort Greene Cumberland.

Since the stay is only for two nights, the minimum nights for | 74
accommodation should be limited to less than 3, and the accommodations further filtered
based on the criterion are: Affordable Private Spacious Room in Brooklyn, Extra Cozy Room
in Center of Williamsburg, Fort Greene Room, Cozy room in Bushwick– 15 min to the city.

Among the filtered accommodations, I should choose the one with the lowest | 75
 price:

Affordable Private Spacious Room in Brooklyn has a maximum | 76
occupancy of 2 people and is priced at $790. Since we are 2 people, we need 1 room, the
total price is $790 * 1 = $790.

Extra Cozy Room in Center of Williamsburg has a maximum occupancy | 77
of 1 people and is priced at $1033. Since we are 2 people, we need 2 room, the total price
 is $1033 * 2 = $2066.

Fort Greene Room has a maximum occupancy of 1 people and is priced | 78
 at $722. Since we are 2 people, we need 2 room, the total price is $722 * 2 = $1444.

Cozy room in Bushwick– 15 min to the city has a maximum occupancy | 79
of 2 people and is priced at $933. Since we are 2 people, we need 1 room, the total price
is $933 * 1 = $933.

Based on the calculations above, I will submit: "Affordable | 80
Private Spacious Room in Brooklyn, Chattanooga".

## Blocksworld:

```
[Plan]                                                                                    1
    [to get the blue block clear]                                                          2
        I can unstack the yellow block from on top of the blue block                       3
        The current state is: the orange block is on the table and not clear, the          4
red block is on top of the orange block and not clear, the blue block is on top of the red
 block and clear, the yellow block is in my hand and not clear.
    [to get the blue block on the table]                                                   5
        I can put down the yellow block.                                                    6
        The current state is: the orange block is on the table and not clear, the          7
red block is on top of the orange block and not clear, the blue block is on top of the red
 block and clear, the yellow block is on the table and clear.
        I can unstack the blue block from on top of the red block.                          8
        The current state is: the orange block is on the table and not clear, the          9
red block is on top of the orange block and clear, the blue block is in my hand and not
clear, the yellow block is on the table and clear.
        I can put down the blue block.                                                      10
        The current state is: the orange block is on the table and not clear, the          11
red block is on top of the orange block and clear, the blue block on the table and clear,
```

```
the yellow block is on the table and clear.                                          
        [to get the orange block clear]                                               12
                I can unstack the red block from on top of the orange block.          13
                The current state is: the orange block is on the table and clear, the red  14
block is in my hand and not clear, the blue block on the table and clear, the yellow block
 is on the table and clear.
        [to get the orange block on top of the blue block]                            15
                I can put down the red block.                                          16
                The current state is: the orange block is on the table and clear, the red  17
block is on the table and clear, the blue block on the table and clear, the yellow block
is on the table and clear.
                I can pick up the orange block.                                        18
                The current state is: the orange block is in my hand and not clear, the  19
red block is on the table and clear, the blue block on the table and clear, the yellow
block is on the table and clear.
                I can stack the orange block on top of the blue block.                 20
                The current state is: the orange block is on top of the blue block and  21
clear, the red block is on the table and clear, the blue block on the table and not clear,
 the yellow block is on the table and clear.
        [to get the red block on top of the orange block]                             22
                I can pick up the red block.                                           23
                The current state is: the orange block is on top of the blue block and  24
clear, the red block is in my hand and not clear, the blue block on the table and not
clear, the yellow block is on the table and clear.
                I can stack the red block on top of the orange block.                  25
                The current state is: the orange block is on top of the blue block and not  26
 clear, the red block is on top of the orange block and clear, the blue block on the table
 and not clear, the yellow block is on the table and clear.
```

## Mystery Blocksworld:

```
[Plan]                                                                                1
        [to get Province object b becomes True]                                       2
                Since Harmony is True, Province object a is True, object a Craves object b  3
 is True, I can Feast object a from object b.
                Province object a, object a Craves object b becomes False              4
                Hamony becomes False                                                  5
                Pain object a, Province object b becomes True.                         6
                Harmony is False, the Pain object is a                                 7
                Province object b is True, object b Craves object c                    8
                object c Craves object d                                               9
                Planet object d is True.                                               10
                The subtask is achieved.                                               11
        [to get Planet object b becomes True]                                         12
                Since Harmony is False, the Pain object is a, I can succumb object a.  13
                Planet object a, Province object a becomes True                        14
                Harmony becomes True                                                   15
                Pain object a becomes False.                                          16
                Planet object a, Province object a is True                             17
                Province object b is True, object b Craves object c                    18
                Province object c is False                                             19
                Planet object d is True.                                               20
                Since Harmony is True, Province object b is True, object b Craves object c  21
 is True, I can Feast object b from object c.
                Province object b, object b Craves object c becomes False              22
                Hamony becomes False                                                  23
                Pain object b, Province object c becomes True.                         24
                Planet object a, Province object a is True                             25
                Province object c is True, object c Craves object d                    26
                Planet object d is True.                                               27
                Since Harmony is False, the Pain object is b, I can succumb object b.  28
                Planet object b, Province object b becomes True                        29
                Harmony becomes True                                                   30
                Pain object b becomes False.                                          31
                Harmony is True                                                        32
```

```
                   Planet object a, Province object a is True                          33
                   Planet object b, Province object b is True                          34
                   Province object c is True, object c Craves object d                 35
                   Planet object d is True.                                            36
                   The subtask is achieved.                                            37
        [to get Province object d becomes True]                                        38
                   Since Harmony is True, Province object c is True, object c Craves object d  39
 is True, I can Feast object c from object d.
                   Province object c, object c Craves object d becomes False           40
                   Hamony becomes False                                                41
                   Pain object c, Province object d becomes True.                       42
                   Planet object a, Province object a is True                          43
                   Planet object b, Province object b is True                          44
                   Planet object d, Province object d is True.                         45
                   The subtask is achieved.                                            46
        [to get Province object b becomes True]                                        47
                   The subtask is achieved.                                            48
        [to get object d Crave object b]                                               49
                   Since Harmony is False, the Pain object is c, I can succumb object c.  50
                   Planet object c, Province object c becomes True                      51
                   Harmony becomes True                                                52
                   Pain object c becomes False.                                        53
                   Planet object a, Province object a is True                          54
                   Planet object b, Province object b is True                          55
                   Planet object c, Province object c is True                          56
                   Province object d, Planet object d is True.                         57
                   Since Harmony is True, Province object d, Planet object d is True, I can  58
Attack object d.
                   Province object d, Planet object d becomes False                    59
                   Harmony becomes False                                               60
                   Pain object d becomes True.                                         61
                   Planet object a, Province object a is True                          62
                   Planet object b, Province object b is True                          63
                   Planet object c, Province object c is True.                         64
                   Since Harmony is False, the Pain object is d, Province object b is True, I  65
 can Overcome object d from object b.
                   Province object d, object d Craves object b becomes True            66
                   Harmony becomes True                                                67
                   Province object b, Pain object d becomes False.                      68
                   Planet object a, Province object a is True                          69
                   Planet object b is True                                             70
                   Planet object c, Province object c is True                          71
                   Province object d is True, object d Craves object b.                72
                   The subtask is achieved.                                            73
        [to get Province object c becomes True]                                        74
                   The subtask is achieved.                                            75
        [to get Province object d becomes True]                                        76
                   The subtask is achieved.                                            77
        [to get object c Crave object d]                                               78
                   Since Harmony is True, Province object c, Planet object c is True, I can  79
Attack object c.
                   Province object c, Planet object c becomes False                    80
                   Harmony becomes False                                               81
                   Pain object c becomes True.                                         82
                   Planet object a, Province object a is True                          83
                   Planet object b is True                                             84
                   Province object d is True, object d Craves object b.                85
                   By [3.2], since Harmony is False, the Pain object is c, Province object d  86
is True, I can Overcome object c from object d.
                   Province object c, object c Craves object d becomes True            87
                   Harmony becomes True                                                88
                   Province object d, Pain object c becomes False.                      89
                   Planet object a, Province object a is True                          90
                   Planet object b is True                                             91
                   Province object c is True, object c Craves object d                 92
```

```
                    object d Craves object b.                                 93
                    The subtask is achieved.                                  94
```

**Trip Planning:**

```
[Plan]                                                                        1
        For the Venice part:                                                  2
        I need to consider two parts: the dates for staying in Venice, and the  3
transportation to the next city.
                To consider the dates for staying in Venice:                  4
                The rest of the time is from day 1 to day 7.                   5
                The required stay duration is 3 days, and it is necessary to be in Venice  6
from day 5 to day 7.
                So I will submit: "**Day 5-7:** Visit Venice for 3 days".      7
                To consider the transportation to plan for next:              8
                According to the flight information, Venice can lead to Berlin, so next we  9
 will think about Berlin.
                Since the stay in Berlin should be connected to the stay in Venice, it  10
should either end on day 5 or start on day 7.
                Since the rest of the time is from day 1 to day 5, the stay in Berlin  11
should end on day 5.
                So the transportaion I will submit is: "**Day 5:** Fly from Berlin to  12
Venice."
        For the Berlin part:                                                  13
        I need to consider two parts: the dates for staying in Berlin, and the  14
transportation to the next city.
                To consider the dates for staying in Berlin:                  15
                The rest of the time is from day 1 to day 5.                   16
                The required stay duration is 4 days, and the stay in Berlin should end on  17
 day 5, so it is necessary to be in Berlin from day 2 to day 5.
                So I will submit: "**Day 2-5:** Visit Berlin for 4 days".      18
                To consider the transportation to plan for next:              19
                According to the flight information, Berlin can lead to Venice and Tallinn  20
. Because Venice has already discussed it, so next we will talk about Tallinn next.
                Since the stay in Tallinn should be connected to the stay in Berlin, it  21
should either end on day 2 or start on day 5.
                Since the rest of the time is from day 1 to day 2, the stay in Tallinn  22
should end on day 2.
                So the transportaion I will submit is: "**Day 2:** Fly from Tallinn to  23
Berlin."
        For the Tallinn part:                                                 24
        I need to consider two parts: the dates for staying in Tallinn, and the  25
transportation to the next city.
                To consider the dates for staying in Tallinn:                 26
                The rest of the time is from day 1 to day 2.                   27
                The required stay duration is 2 days, and the stay in Tallinn should end  28
on day 2, so it is necessary to be in Tallinn from day 1 to day 2.
                Notice that day 1 is in the dates for staying in Tallinn, so need to  29
arrive in Tallinn and visit.
                I will submit: "**Day 1-2:** Arriving in Tallinn and visit Tallinn for 2  30
days".
                To consider the transportation to plan for next:             31
                Since the rest of the time is empty, the trip plan has been completed.  32
        Based on the above plan, I need to sort them in time, and use flights to connect  33
these cities for trip plan.
        According to the chronological order, day 1 to day 2 is Tallinn, day 2 to day 5 is  34
 Berlin, and day 5 to day 7 is Venice.
        So I will fly from Tallinn to Berlin in day2, and fly from Berlin to Venice in  35
day5.
```

## E.4. Examples of the Final Plan

**TravelPlanner:**

```
Travel Plan:                                                                    1
Day 1:                                                                          2
Current City: from Houston to Nashville                                         3
Transportation: Flight Number: F3956409, from Houston to Nashville, Departure Time: 17:36,   4
 Arrival Time: 19:12
Breakfast: -                                                                    5
Attraction: Country Music Hall of Fame and Museum, Nashville                    6
Lunch: -                                                                        7
Dinner: Twigly, Nashville                                                       8
Accommodation: Lovely room in heart of Williamsburg, Nashville                  9
                                                                                10
Day 2:                                                                          11
Current City: Nashville                                                         12
Transportation: -                                                               13
Breakfast: Bablu Fast Food, Nashville                                           14
Attraction: Nashville Zoo at Grassmere, Nashville                               15
Lunch: Kitchen King, Nashville                                                  16
Dinner: Govinda's Confectionery, Nashville                                      17
Accommodation: Lovely room in heart of Williamsburg, Nashville                  18
                                                                                19
Day 3:                                                                          20
Current City: from Nashville to Knoxville                                       21
Transportation: taxi, from Nashville to Knoxville, duration: 2 hours 42 mins    22
Breakfast: Biryani By Kilo, Knoxville                                           23
Attraction: World's Fair Park, Knoxville.                                       24
Lunch: Open Kitchen, Knoxville                                                  25
Dinner: Chit Chat, Knoxville                                                    26
Accommodation: Light-filled Room in Renovated Apt, Knoxville                    27
                                                                                28
Day 4:                                                                          29
Current City: Knoxville                                                         30
Transportation: -                                                               31
Breakfast: Mamagoto, Knoxville                                                  32
Attraction: Knoxville Museum of Art, Knoxville                                  33
Lunch: La-Nawaab, Knoxville                                                     34
Dinner: Tandoori Tadka, Knoxville                                               35
Accommodation: Light-filled Room in Renovated Apt, Knoxville                    36
                                                                                37
Day 5:                                                                          38
Current City: from Knoxville to Chattanooga                                     39
Transportation: taxi, from Knoxville to Chattanooga, duration: 1 hour 41 mins   40
Breakfast: Tpot, Chattanooga                                                    41
Attraction: The Chattanooga Zoo at Warner Park, Chattanooga.                    42
Lunch: Liquid, Chattanooga                                                      43
Dinner: Muradabadi, Chattanooga                                                 44
Accommodation: Affordable Private Spacious Room in Brooklyn, Chattanooga        45
                                                                                46
Day 6:                                                                          47
Current City: Chattanooga                                                       48
Transportation: -                                                               49
Breakfast: Burger's King, Chattanooga                                           50
Attraction: Rock City Gardens, Chattanooga.                                     51
Lunch: Basil Tree, Chattanooga                                                  52
Dinner: Sardar A Pure Meat Shop, Chattanooga                                    53
Accommodation: Affordable Private Spacious Room in Brooklyn, Chattanooga        54
                                                                                55
Day 7:                                                                          56
Current City: from Chattanooga to Houston                                       57
Transportation: taxi, from Chattanooga to Houston, duration: 11 hours 47 mins   58
Breakfast: Pizza Hut Delivery, Chattanooga                                      59
Attraction: Tennessee Aquarium, Chattanooga.                                    60
Lunch: -                                                                        61
Dinner: -                                                                       62
Accommodation: -                                                                63
```

**Blocksworld:**

```
[PLAN]                                                                          1
unstack the yellow block from on top of the blue block                          2
put down the yellow block                                                       3
unstack the blue block from on top of the red block                             4
put down the blue block                                                         5
unstack the red block from on top of the orange block                           6
put down the red block                                                          7
pick up the orange block                                                        8
stack the orange block on top of the blue block                                 9
pick up the red block                                                           10
stack the red block on top of the orange block                                  11
[PLAN END]                                                                      12
```

**Mystery Blocksworld:**

```
[PLAN]                                                                          1
feast object a from object b                                                    2
succumb object a                                                                3
feast object b from object c                                                    4
succumb object b                                                                5
feast object c from object d                                                    6
succumb object c                                                                7
attack object d                                                                 8
overcome object d from object b                                                 9
attack object c                                                                 10
overcome object c from object d                                                 11
[PLAN END]                                                                      12
```

**Trip Planning:**

```
Trip Plan:                                                                      1
**Day 1-2:** Arriving in Tallinn and visit Tallinn for 2 days.                  2
**Day 2:** Fly from Tallinn to Berlin.                                          3
**Day 2-5:** Visit Berlin for 4 days.                                           4
**Day 5:** Fly from Berlin to Venice.                                           5
**Day 5-7:** Visit Venice for 3 days.                                           6
```

