# OpenReview forum: "HyperTree Planning: Enhancing LLM Reasoning via Hierarchical Thinking"
_ICML.cc/2025/Conference — ICML 2025 poster_

### Official Review · Reviewer_uus3 · 2025-03-09

**Overall Recommendation:** 4

**Summary:**

This paper proposes a tree-based planning strategy, called **HyperTree Planning (HTP)**, which utilizes a hypertree-structured planning framework. HTP adopts a divide-and-conquer approach, decomposing a complex goal into several sub-goals in a top-down manner. These sub-goals are further decomposed iteratively until they are indivisible. The authors define this process in four stages: selection, expansion, construction, and decision, which together form a hyper-tree structure for further planning. During the plan generation stage, HTP accesses knowledge bases to refine the plan outline, aiming to generate a comprehensive and detailed plan rather than just an outline. The experiments, covering three fundamental models and three widely-used planning benchmarks, demonstrate the effectiveness of HTP. Additionally, ablation studies further validate the proposed strategies.

**Claims And Evidence:**

Yes, all the claims made in the submission are supported by experiments.

**Essential References Not Discussed:**

The discussion of related work is sufficient, and the experiments cover the necessary baselines.

**Experimental Designs Or Analyses:**

I believe the experimental design is comprehensive and convincing. The authors validate the effectiveness of their proposed methodology across three foundational models and three planning benchmarks, comparing it to seven previous works. All the experiments demonstrate the superiority of their proposed method. Overall, the soundness and validity of the experiments in this paper are strong.

**Methods And Evaluation Criteria:**

The proposed method makes sense, and the evaluation criteria are adopted from the official benchmarks' criteria.

**Other Comments Or Suggestions:**

Please refer to the weaknesses mentioned in the last part.

**Other Strengths And Weaknesses:**

In my opinion, this paper is strong, and I have highlighted the strengths in the above parts. However, I do have some concerns regarding the strategy:

1. **Cost concerns**: As shown in Table 2, the cost for TravelPlanner is nearly 3x higher compared to vanilla GPT-4 and Gemini-1.5-Pro, with the issue becoming even more prominent in PlanBench (\~8x) and Natural Plan (\~14x). While these costs are still lower than those of the o1-preview, the large cost disparity on the same foundation model remains a concern. I acknowledge that this is a common issue with all tree-based methodologies because of the huge search space, and I encourage the authors to include more discussion on this matter, as well as potential improvements.

2. **Lack of bad case studies**: Another concern is the apparent lack of bad case studies. There is still a significant gap compared to the saturated performance on the three planning benchmarks. I wonder which stage(s) primarily contribute to the failure cases and why. Are these failures due to the current LLMs' inability to identify potential errors, or is it something else? A detailed analysis of these issues would benefit future work.

By the way, I believe the current version meets the standard for ICML, and I consider this a strong paper. However, it would benefit greatly from addressing the aforementioned concerns.

**Questions For Authors:**

1. In the "Selection" stage, the authors mention that they primarily rely on LLM-based evaluation, both for the selection of hyperchains and leaf nodes. Would it be more effective to use training-based process reward models or other model-based evaluators? While the current strategy is certainly more generalizable, could it be improved by incorporating such methods?

**Relation To Broader Scientific Literature:**

This paper introduces a novel hypertree-based planning strategy that enhances the performance of LLMs on planning tasks. Both the comprehensive studies and additional analyses demonstrate the effectiveness of the proposed methodology. Furthermore, the authors show that the proposed HTP yields more significant performance improvements on tasks with longer reasoning chains, which is particularly impressive and strongly validates its promising future potential.

**Theoretical Claims:**

There is no proof for a theoretical claim in this paper.

---

> ### Author Rebuttal · Authors · 2025-03-31
>
> We thank the reviewer for the insightful, valuable, and positive comments. We address the concerns in detail as follows. We sincerely hope that our response could properly address your concerns.
> ### Weakness 1
> >discussion on cost concern and potential improvements.
>
> We will incorporate **a discussion of computational cost** in the revised version of our paper, as detailed below:
>
> - The inference cost of HTP can be higher than conventional backbone models, primarily due to two factors: (1) **the computational cost introduced by multi-path reasoning** (i.e., tree search) and (2) **the hierarchical reasoning mechanism, which decomposes the original task into subproblems, leading to more detailed reasoning**.
> - **The impact of these factors varies across tasks**. In TravelPlanner, the additional cost mainly comes from the need for detailed reasoning to satisfy constraints (factor 2). In Natural Plan, where we need to find a unique solution through trial and error, tree search (factor 1) is the primary source of overhead.
> - Despite these additional costs, **HTP remains more computationally efficient than MCTS-based methods**. We have added additional experimental results and explanations on this aspect; please refer to **Weaknesses 1 of Reviewer rT6H**.
> - We propose two potential improvements: (1) **Using fine-tuned small models instead of large models during the hypertree construction stage**, which could reduce the computational burden; (2) **Predicting task complexity through meta-learning and dynamically adjusting hypertree parameters** (e.g., depth, width) during hypertree construction, allowing for more efficient resource allocation.
> ### Weakness 2
> >Lack of bad case studies for future work
>
> We will incorporate **an analysis of bad cases** in our revised paper, as detailed below:
> - **LLMs struggle with complex single-step reasoning**. In TravelPlanner, given a table of candidate hotels (including hotel name, room type, etc.), the model is required to identify 'entire room' options. However, it often fails by missing correct options or selecting incorrect ones.
> - **LLMs lack human prior knowledge**. In TravelPlanner, humans naturally use strategies to stay within budget, such as choosing self-driving over flights, selecting the cheapest hotels and restaurants. However, models struggle to leverage such prior knowledge, resulting in **plans that exceed the budget**.
> - **HTP remains vulnerable to long-horizon errors**. In PlanBench, the model must track and update the environment state after each action. However, any mistake in state updating can propagate errors, leading to incorrect subsequent reasoning and actions.
> - **HTP lacks the capability for self-reflection and backtracking**. In Natural Plan, the model must determine a feasible multi-city route. While HTP outperforms baselines, it may struggle with constraint violations, such as selecting consecutive cities without a direct connection. Humans typically backtrack and revise their decisions in such cases, whereas HTP lacks an inherent mechanism for such adjustments.
>
> Based on the analysis, we have added **a discussion on future work**; please refer to **Weakness 5 of Reviewer zPJM**.
> ### Question 1
> >In the "Selection" stage, would it be more effective to use training-based process reward models or other model-based evaluators?
>
> We absolutely agree that incorporating a reward-based approach could enhance decision accuracy, though it comes at the cost of some generalizability. To investigate this, **we conduct experiments on the Blocksworld task** from the PlanBench dataset, which is particularly challenging because it requires selecting the correct actions, making the **selection stage highly influential on overall performance**. We design two types of reward functions, and all methods use GPT-4o as the backbone model:
> - **Rule-based rewards**: Rewards explicitly defined based on predefined heuristics, providing structured guidance for action selection.
> - **LLM-based rewards**: Rewards generated dynamically by prompting LLMs to formulate heuristic functions, reducing the reliance on manual reward design. The rewards are further refined through an evolutionary algorithm, iteratively improving the effectiveness.
>
> Our results show that **even simple rule-based rewards can significantly improve model performance**. Moreover, the **LLM-generated heuristic rewards offer more fine-grained guidance while reducing the burden of manual reward engineering**. We believe this is a promising direction for future research.
> |Method|HTP|+Rule-based|+LLM-based|
> |-|:-:|:-:|:-:|
> |**Success Rate**|**54.7**|**59.7**|**71.2**|

---

> > ### Comment · Reviewer_uus3 · 2025-04-02
> >
> > Thanks for your elaboration and complementary experiments. I have no further questions and will maintain my score.

---

> > > ### Author Response · Authors · 2025-04-02
> > >
> > > Thank you very much for your thoughtful review. We deeply appreciate your constructive suggestions, which have significantly contributed to strengthening our work.
> > >
> > > Your insights on computational cost, failure case analysis, and selection module enhancement are highly valuable. We will ensure that these improvements are incorporated into the revised manuscript.

---

### Official Review · Reviewer_zPJM · 2025-03-11

**Overall Recommendation:** 3

**Summary:**

This paper concentrates on complex planning tasks, for instance, mathematical and logical reasoning. To alleviate the multiple challenges, extended reasoning steps, diverse constraints, and multiple distinct sub-tasks, they propose a HyperTree Planning (HTP) that is based on the divide-and-conquer strategy to split tasks into multiple distinct sub-tasks. Extensive experiments demonstrated the proposed method improved the accuracy of the TravelPlanner benchmark.

**Claims And Evidence:**

Yes.

**Essential References Not Discussed:**

No

**Experimental Designs Or Analyses:**

Yes. They develop multiple experiments and adopt multiple baselines, planning strategies, in-context learning methods, and powerful LLMs to support the effectiveness of the proposed HTP.

**Methods And Evaluation Criteria:**

Yes.

**Other Comments Or Suggestions:**

There lack of limitations and future work.

**Other Strengths And Weaknesses:**

**Strengths**

The paper is the first work to combine hypertree structure into a reasoning process.

To alleviate the fluctuation of human-designed prompts, they design an autonomous planning framework that self-guide the planning process without manual design.

They conduct amount of experiments to support the effectiveness of proposed methods.

**Weakness**

I am interested how the ability of humans in these involved benchmarks, Travel Planner, PlanBench, and Natural Plan. Whether humans almost not make errors across multiple complexity tasks?

These should provide some failure cases to analyze the limitations of current methods.

Though they conduct a number of experiments to investigate the capability of proposed methods, the analysis mainly includes describing the result with explicit performance. There lack of deep analysis of why these methods improve the performance compared to related works. Furthermore, if existing methods remain a great gap between humans, why attributes to this pattern?

The author should discuss how the future works utilize this work and any potential value in multiple real-world tasks.

If the author provides a reasonable response, I will reconsider my ranking.

**Questions For Authors:**

See above.

**Relation To Broader Scientific Literature:**

The earliest works begin with analogical reasoning, including CoT, and ToT.
Most recently, the agent systems collaborate through structured processes to combat planning tasks.
The author proposes the existing limitations in complex task reasoning include concentrating on mathematical and logical reasoning that is ill-suited for planning questions, the performance depends on human-designed prompts, and the generalization across tasks hindered by human-designed intervention in autonomous agent methods. They then devise methods that combine ToT and HyperChain to solve these challenges.

**Theoretical Claims:**

Yes. The method has comparatively less proof and no potential questions.

---

> ### Author Rebuttal · Authors · 2025-03-30
>
> We thank the reviewer for the insightful, valuable, and positive comments. We address the concerns in detail as follows. We sincerely hope that our response could properly address your concerns.
> ### Weakness 1
> >Do humans almost never make errors across multiple complex tasks?
>
> In **TravelPlanner**, humans can achieve **near-perfect accuracy**. For PlanBench and Natural Plan, human performance data is unavailable. We estimate that humans can achieve **near-perfect accuracy in PlanBench**, as stacking blocks aligns well with human capabilities. In **Natural Plan**, we expect an overall success rate of **approximately 80%**, with errors occurring mainly on the most difficult tasks.
> ### Weakness 2 & Weakness 4
> >provide some failure cases to analyze the limitations of current methods
>
> We will incorporate **an analysis of bad cases** in our revised paper, as detailed below:
> - **LLMs struggle with complex single-step reasoning**. In TravelPlanner, given a table of candidate hotels (including hotel name, room type, etc.), the model is required to identify 'entire room' options. However, it often fails by missing correct options or selecting incorrect ones.
> - **LLMs lack human prior knowledge**. In TravelPlanner, humans naturally use strategies to stay within budget, such as choosing self-driving over flights, selecting the cheapest hotels and restaurants. However, models struggle to leverage such prior knowledge, resulting in **plans that exceed the budget**.
> - **HTP remains vulnerable to long-horizon errors**. In PlanBench, the model must track and update the environment state after each action. However, any mistake in state updating can propagate errors, leading to incorrect subsequent reasoning and actions.
> - **HTP lacks the capability for self-reflection and backtracking**. In Natural Plan, the model must determine a feasible multi-city route. While HTP outperforms baselines, it may struggle with constraint violations, like selecting consecutive cities without a direct connection. Humans typically backtrack and revise their decisions in such cases, whereas HTP lacks an inherent mechanism for such adjustments.
> >If existing methods remain a great gap between humans, why attributes to this pattern?
>
> A considerable gap remains between HTP and human-level planning, which can be mainly attributed to:
> - **Humans are adept at planning and adapting to complex constraints**, such as effortlessly filtering hotels based on multiple conditions.
> - **Humans leverage extensive prior knowledge beyond what is provided in the prompt**, which helps them make more informed, practical decisions (e.g., cost-saving strategies in travel planning).
> - **Humans are less prone to hallucinations**, making them more reliable for long-horizon planning.
> - **Humans can self-reflect and backtrack when necessary**, adjusting their decisions in the face of contradictions.
> ### Weakness 3
> >Why HTP improve the performance compared to related works?
>
> Please refer to **Question 1 of Reviewer rT6H**.
> ### Weakness 5
> >How the future works utilize this work and any potential value in multiple real-world tasks?
>
> We will incorporate **a discussion on future work and potential applications** in our revised paper, as detailed below:
>
> **First, HTP naturally integrates with self-reflection and backtracking techniques, making it particularly valuable for real-world tasks like meeting planning and calendar scheduling**. By replacing single-chain reasoning with a hierarchical hyperchain structure, HTP enhances reflection accuracy by **directly identifying and correcting errors without revisiting unrelated paths**. We conduct **a preliminary experiment** on the TravelPlanner dataset, using GPT-4o as the backbone model. We use a heuristic oracle to generate reflection signals and employ the simple cosine similarity metric to pinpoint specific erroneous paths, limiting reflections to 3 per plan. **Results show significant improvements with reflection, highlighting HTP's strong potential when integrated with reflection-based methods**. Future work can explore two directions: **(1) replacing the heuristic oracle with an LLM-powered self-reflection mechanism to enhance adaptability, and (2) refining the accuracy of error localization to further improve efficiency**.
> ||CPR (Micro)|CPR (Macro)|HCPR (Micro)|HCPR (Macro)|SR|
> |-|:-:|:-:|:-:|:-:|:-:|
> |HTP|87.2|37.8|44.3|32.2|20.0|
> |**HTP+reflection**|**93.6**|**66.7**|**60.7**|**48.9**|**44.4**|
>
> **Second, HTP shows great potential in autonomous agent decision-making due to its scalability and adaptability**. To further explore this, we have added **two agent experiments**; please refer to **Question 2 of Reviewer X7uh**. Future work can explore how to **empower end-to-end agents with autonomous hierarchical thinking using HTP**.
>
> **Third, combining HTP with LLM-based heuristic process reward functions is a promising direction**. We have added **an experiment on this aspect**; please refer to **Question 1 of Reviewer uus3**.

---

### Official Review · Reviewer_rT6H · 2025-03-11

**Overall Recommendation:** 3

**Summary:**

This paper introduces HyperTree Planning (HTP), a reasoning paradigm designed to improve complex planning tasks using hierarchical hypertree-structure.

**Claims And Evidence:**

The core motivation is that existing reasoning methods (e.g., CoT, ToT) struggle with long-horizon, multi-constraint planning problems, such as travel planning, which require handling interdependent sub-tasks. The proposed framework aims to overcome the limitations of intensive human interventions as in existing planning agents. Key contributions in methodology:
* HyperTree Reasoning Paradigm: it adopts a hypertree structure to model the reasoning process to perform hierarchical thinking. The proposed hypertree construction algorithm aligns well with the structured nature of planning problems.
* Autonomous Planning Framework: leverages task-specific planning outlines to self-guide the planning process dynamically, reducing reliance on manually crafted in-context learning examples.

**Essential References Not Discussed:**

Main related work has been included.

**Experimental Designs Or Analyses:**

Overall, the empirical studies show promising performance improvement on benchmark datasets compared to different baseline planning algorithms.

Here are some questions:
1. While a comparison with ReAct is provided in Appendice C.2, why the authors did not compare with ReAct in the main result of Table 1?

**Methods And Evaluation Criteria:**

The proposed HTP framework introduced in this paper utilizes the divide-and-conquer strategy to construct and refine hypertree-based planning outlines and thus organize sub-tasks in a structured manner. It naturally models complex planning tasks. The iterative refinement strategy is also reasonable.

**Other Comments Or Suggestions:**

Please see above.

**Other Strengths And Weaknesses:**

While hypertree reasoning is conceptually appealing, its computational cost relative to existing planning strategies is unclear. It is beneficial to include further analyses in terms of efficiency metrics, e.g. inference speed, memory usage, computational complexity etc.

**Questions For Authors:**

1. While I understand that HyperTree Planning differs from Tree of Thought and RAP by the fact that it constructs hypertree, I cannot fully grasp why the proposed HyperTree Planning excels existing tree-based methods in principle. Say, is it because hypertree structure can better handle planning constraints while ToT and RAP cannot etc. Can authors further comment and compare?

2. Methodologically, why HyperTree Planning can work better than ReAct? ReAct is also able to break tasks into subtasks and iteratively making decisions based on evaluation results.

**Relation To Broader Scientific Literature:**

Authors provide a hypertree-based planning framework for complex planning tasks. It can be helpful for LLM reasoning and agentic frameworks in practice.

**Theoretical Claims:**

This is an empirical work, no theory results.

---

> ### Author Rebuttal · Authors · 2025-03-30
>
> We thank the reviewer for the insightful, valuable, and positive comments. We address the concerns in detail as follows. We sincerely hope that our response could properly address your concerns.
> ### Weaknesses 1
> >It is beneficial to include further analyses in terms of efficiency metrics.
>
> We will incorporate **an analysis of computational cost** in the revised version of our paper, as detailed below. Specifically, we evaluate the computational cost of CoT, RAP, and our HTP on the TravelPlanner dataset. Since open-source models currently exhibit suboptimal performance on TravelPlanner, we adopt GPT-4o as the backbone model. Our evaluation focuses on three commonly used efficiency metrics: **inference speed, token cost, and computational complexity**. Let $n$ denote the number of branches expanded at each step ($n\leq 2$ in TravelPlanner), $l$ the average number of reasoning steps per chain, and $k$ the sampling trajectories in MCTS-based methods.
> |Model|Inference Speed (s)|Token Cost (in/out)|Computational Complexity|
> |:-:|:-:|:-:|:-:|
> |CoT|6.92|4328/641|$O(l)$|
> |RAP|41.94|5440/3374|$O(nkl)$|
> |**HTP**|**25.27**|**5562/963**|$O(nl)$|
>
> The results indicate that **HTP achieves significantly lower computational costs across all three metrics compared to RAP**. This is because our top-down approach eliminates the need for MCTS, reducing the associated overhead.
>
> Moreover, **HTP does not introduce additional computational complexity in hierarchical reasoning** within the hyperchain structure. While the hyperchain structure increases the dimensionality of reasoning, it simultaneously shortens the average path length, allowing the overall computational efficiency to remain at the same level. Additionally, we provide **a comparison of API costs in Table 2 of our paper** for reference.
>
> Besides the experiments, we provide **additional discussions on computational cost**; please refer to **Weakness 1 of Reviewer uus3**.
> ### Question 1
> >Why HTP excels existing tree-based methods in principle?
>
> As illustrated in Figure 2, existing tree-based reasoning methods, such as ToT and RAP, fundamentally select a single reasoning chain from multiple candidate thought chains. **This single-chain structure inherently limits their ability to address complex, long-horizon planning tasks, as it lacks hierarchy and fails to handle multiple subtasks independently.** As a result, these methods often suffer from issues such as **unmet constraints and missing components**.
>
> In contrast, **HTP replaces the single-chain structure with hyperchains, enabling hierarchical reasoning by allowing a single edge to connect multiple nodes**. In travel planning, where about 15 different constraints should be considered, this structure enables the model to clearly identify which subtask needs to be handled along each path, thereby guiding it to **reason about relevant constraints while avoiding unnecessary ones**. Furthermore, **by explicitly decomposing subtasks across multiple layers, each sub-task is represented as an independent node from the outset**. This ensures that **HTP enforces completeness**, preventing essential sub-goals from being omitted.
> ### Question 2
> >Why the authors did not compare with ReAct in the main result of Table 1?
>
> As mentioned in Appendix C.2, the TravelPlanner dataset consists of a two-stage mode (TS) and a sole-planning mode (SP). The TS mode retrieves target information via tool calls before reasoning (similar to ReAct), whereas the SP mode provides all information upfront, requiring the model to filter relevant details during reasoning. **As Table 1 reports baseline results under the SP mode, ReAct is not included**.
>
> To provide a clearer comparison with ReAct, we have conducted **additional experiments evaluating HTP in the TS mode**, and these results will be included in the revised version of our paper. In HTP’s Self-Guided Planning module, instead of retrieving information from context, we modified it to retrieve information via tool calls. All methods use GPT-4o as the backbone model, and the evaluation metrics remain the same as in Table 1. **The results indicate that HTP in the TS mode is comparable to its performance in the SP mode and significantly outperforms ReAct.**
> ||CPR (Micro)|CPR (Macro)|HCPR (Micro)|HCPR (Macro)|SR|
> |-|:-:|:-:|:-:|:-:|:-:|
> |React|79.4|8.33|7.14|5.00|1.67|
> |**HTP(TS)**|83.7|**33.9**|39.5|**28.9**|**18.3**|
> |**HTP(SP)**|87.2|**37.8**|44.3|**32.2**|**20.0**|
> > Why HyperTree Planning can work better than ReAct?
>
> ReAct indeed follows an iterative decision-making process and performs decomposition implicitly through step-by-step reasoning. However, similar to ToT and RAP, **ReAct still follows a single-chain reasoning structure, leading to a lack of hierarchy, unmet constraints, and missing components**. In contrast, HTP overcomes these challenges by leveraging hyperchains to enable structured, hierarchical reasoning, ensuring comprehensive and constraint-aware planning.

---

### Official Review · Reviewer_X7uh · 2025-03-14

**Overall Recommendation:** 3

**Summary:**

This paper proposes an autonomous planning framework called HyperTree Planning that involves (a) HyperTree Constrution, (b) Self-Guided Planning and (c) Plan Generation. It tackes the limitation of exisiting chain-of-thought and tree-of-thought on planning problems, for example, they focus on mathematical and logical reasoning. The main contribution comes from its HyperTree planning on top of existing Tree-of-though, with four steps: (1) Selection (2) Expansion (3) Construction and (4) Decision. Through experiments on travel planning benchmarks, it achieves state-of-the-art performanc and shows compatibility with multiple backbone LLMs.

**Claims And Evidence:**

HyperTree Reasoning shows superior performance on complex planning benchmarks, this claim could be questionable on whether it can be applied to other planning/agent tasks.

**Essential References Not Discussed:**

Not a concern.

**Experimental Designs Or Analyses:**

Yes, I read the main experiment and ablation designs.

**Methods And Evaluation Criteria:**

I think HyperTree Planning shows promising performance on trip planning tasks. While it is a concern whether this framework generalize to other agent tasks such as webArena. HyperTree Reasoning Paradigm:

**Other Comments Or Suggestions:**

N/A

**Other Strengths And Weaknesses:**

The novelty of the paper seems to be limited as a combination of ToT and HyperTree structured proposed in (Lample et al., 2022)

**Questions For Authors:**

1. How is the probability pruning implemented? I assume no probability is generated during the hypertree construction stage.

2. Can HyperPlanning be used in Agent tasks? Especially with the function calling capablity.

**Relation To Broader Scientific Literature:**

This work might be applicable to other Agent datasets such as VisualAgent and WebArena.

**Theoretical Claims:**

There is no theory in the paper.

---

> ### Author Rebuttal · Authors · 2025-04-01
>
> We thank the reviewer for the insightful, valuable, and positive comments. We address the concerns in detail as follows. We sincerely hope that our response could properly address your concerns. If so, we would deeply appreciate it if you could raise your score. If not, please let us know your further concerns, and we will continue actively responding to your comments and improving our submission.
> ### Weakness 1
> >The novelty of the paper seems to be limited as a combination of ToT and HyperTree structured proposed in (Lample et al., 2022)
>
> The novelty compared to "ToT+HyperTree":
> - **Different Motivation**: ToT focuses on exploring the search space through tree search strategies, addressing problems that require extensive trial and error to reach a solution. HyperTree Proof (Lample et al., 2022) is designed for an entirely different domain, aiming to find valid proof pathways by selecting appropriate proof methods. **In contrast, HTP is motivated by the key challenges in complex planning tasks**, such as **a lack of hierarchy, unmet constraints, and missing components**. We take inspiration from human problem-solving strategies and leverage the hypertree structure to structurally enable LLMs to perform hierarchical reasoning. Neither ToT nor HyperTree Proof can effectively address complex planning tasks.
> - **Innovative Usage of HyperTree Structure**: In HyperTree Proof, the hypertree structure is primarily used to **select the correct edges**, as each edge represents a different proof method, and the key goal is to complete the proof process by making the right selections at each layer. In contrast, HTP leverages the hypertree structure to **generate nodes**, enabling a divide-and-conquer strategy that decomposes complex planning tasks into hierarchical subtasks. **While both methods utilize hypertree structures, the modeling and usage are fundamentally different**. Therefore, HTP is not a simple adaptation or combination of existing methods.
> - **Explainability**: We explain **why HTP is particularly well-suited for solving complex planning tasks**, providing a solid rationale for its superior experimental performance. For more details, please refer to **Question 1 of reviewer rT6H**.
> - **Novel Planning Framework**: We innovatively model planning process using the hypertree structure, introducing the first hypertree-based reasoning algorithm and a corresponding fully autonomous planning framework.
> - **Flexibility and Scalability**: Our HTP framework is highly flexible and can be seamlessly integrated with mechanisms such as **self-reflection** and **process reward modeling**. We have conducted **supplementary experiments** on these mechanisms, as detailed in **Weakness 5 of reviewer zPJM** and **Question 1 of reviewer uus3**. This adaptability is crucial for the future development of **autonomous LLM agents**, highlighting HTP’s strong scalability and generalization capabilities.
> ### Question 1
> >How is the probability pruning implemented?
>
> The probabilities in this context refer to **confidence scores outputted by LLMs**, a technique adopted in several recent LLM-based works like RAP (Hao et al., 2023) and Self-DC (Wang et al., 2024). Specifically, to select hyperchains, we enumerate candidates and present them to the LLM in a dictionary format. **The LLM selects a hyperchain by outputting its index along with the corresponding logit, which we exponentiate to obtain the confidence probability**. Since our hypertree structure involves multi-layer selections, the probabilities for different layers within the same hyperchain follow the multiplication rule, ensuring that the total probability sum across all hyperchains remains normalized. This approach enables us to **select the top-n hyperchains with the highest probabilities efficiently while pruning lower-probability candidates**.
> ### Question 2
> >Can HTP be used in Agent tasks? Especially with the function calling capability.
>
> We will supplement **experiments on agent tasks** in the revised version of our paper, as detailed below.
>
> Specifically, we select two widely adopted agent benchmarks, **WebShop and WebArena**, to evaluate HTP’s capability in function calling. As baselines, we choose ReAct, Reflexion, and LATS, which represent SOTA methods without incorporating prior knowledge. To ensure consistency with the baselines, we use GPT-4o and Gemini-1.5-Pro as the backbone models for WebShop and WebArena, respectively. Both datasets are evaluated using **Success Rate (SR)** as the metric.
> |Method|WebShop|WebArena|
> |-|:-:|:-:|
> |React|33.2|17.9|
> |Reflexion|39.8|20.2|
> |LATS|41.0|21.0|
> |**HTP**|**44.2**|**23.4**|
>
> **The results show that HTP outperforms SOTA methods on both datasets, demonstrating its strong performance in function calling scenarios**. Additionally, we provide an extended experiment of **HTP’s function calling performance on the TravelPlanner dataset**; please refer to **Question 2 of Reviewer rT6H**.

---

### Decision · Program_Chairs · 2025-05-01

**Decision:**

Accept (poster)

**Comment:**

The paper reports a study on LLM planning, and in particular, it proposes to apply the hypertree structure to guide complex planning tasks for LLMs. This is a very important research area, and the proposed solution of hypertree planning provides a reasonable solution choice to the community. The reviewers acknowledge the importance and the contribution of the paper, and are satisfied by the authors' replies. Therefore, I would like to recommend acceptance to the paper.